# Recent Advances in Photocatalytic Oxidation of Methane to Methanol

**DOI:** 10.3390/molecules27175496

**Published:** 2022-08-26

**Authors:** Gita Yuniar, Wibawa Hendra Saputera, Dwiwahju Sasongko, Rino R. Mukti, Jenny Rizkiana, Hary Devianto

**Affiliations:** 1Research Group on Energy and Chemical Engineering Processing System, Department of Chemical Engineering, Faculty of Industrial Technology, Institut Teknologi Bandung, Jl. Ganesha No. 10, Bandung 40132, Indonesia; 2Center for Catalysis and Reaction Engineering, Institut Teknologi Bandung, Jl. Ganesha No. 10, Bandung 40132, Indonesia; 3Research Center for New and Renewable Energy, Institut Teknologi Bandung, Jl. Ganesha No. 10, Bandung 40132, Indonesia; 4Division of Inorganic and Physical Chemistry, Faculty of Mathematics and Natural Sciences, Institut Teknologi Bandung, Jl. Ganesha No. 10, Bandung 40132, Indonesia; 5Research Center for Nanoscience and Nanotechnology, Institut Teknologi Bandung, Jl. Ganesha No. 10, Bandung 40132, Indonesia

**Keywords:** photocatalysis, methane oxidation, methanol, light, catalyst

## Abstract

Methane is one of the promising alternatives to non-renewable petroleum resources since it can be transformed into added-value hydrocarbon feedstocks through suitable reactions. The conversion of methane to methanol with a higher chemical value has recently attracted much attention. The selective oxidation of methane to methanol is often considered a “holy grail” reaction in catalysis. However, methanol production through the thermal catalytic process is thermodynamically and economically unfavorable due to its high energy consumption, low catalyst stability, and complex reactor maintenance. Photocatalytic technology offers great potential to carry out unfavorable reactions under mild conditions. Many in-depth studies have been carried out on the photocatalytic conversion of methane to methanol. This review will comprehensively provide recent progress in the photocatalytic oxidation of methane to methanol based on materials and engineering perspectives. Several aspects are considered, such as the type of semiconductor-based photocatalyst (tungsten, titania, zinc, etc.), structure modification of photocatalyst (doping, heterojunction, surface modification, crystal facet re-arrangement, and electron scavenger), factors affecting the reaction process (physiochemical characteristic of photocatalyst, operational condition, and reactor configuration), and briefly proposed reaction mechanism. Analysis of existing challenges and recommendations for the future development of photocatalytic technology for methane to methanol conversion is also highlighted.

## 1. Introduction

Energy plays a vital role in ensuring national security. It becomes a concern because the demand is growing rapidly along with economic development, population growth, and technological development. Currently, energy resources rely on non-renewable energy, such as fossil fuels (coal, natural gas, and oil). Meanwhile, the availability of these resources is finite and will be vanished if they continue to be exploited unwisely. On the other hand, fossil fuel combustion produces CO_2_ that will exacerbate global warming.

The global warming issue has attracted a lot of attention. Through the 2015 Paris Agreement, many countries have committed to contributing to climate change mitigation and reducing greenhouse gas emissions. According to Environmental Protection Agency (EPA), methane is the second-largest contributor to climate change accounting for about 10% of all greenhouse gasses emitted in 2019. The Intergovernmental Panel on Climate Change (IPCC) states that comparatively, it has 34 times greater Global Warming Potential (GWP) than carbon dioxide (Figure 1). Methane is emitted from natural sources (wetlands, termites, marine, freshwater, and CH_4_ hydrate) and human activities (energy, industry, agriculture, land use, and waste management). Globally, 50–65 percent of total methane in the atmosphere emits from human activities.

Methane is the simplest hydrocarbon with a symmetric tetrahedral structure leading methane to have low polarizability and low electron and proton affinity [2]. Methane is a very stable and non-reactive substance. Therefore, it is difficult to activate the C-H bond; hence, high pressure and temperature are required. As the main component of natural gas, methane has a high caloric value and is classified as a good hydrogen source. It is converted into various chemicals such as syngas for ammonia production, methanol, hydrocarbon, acetylene, and carbon disulfide. It can also be used directly as fuel by direct combustion (Figure 2).

Methanol and ammonia are the most critical methane conversion products required in large quantities [3]. Liquefaction of methane to methanol is preferred over gaseous methane because the liquid state is easier to handle, store, and transport. Methanol is also vital in the industry since it is used as feedstock for textiles, pharmaceuticals, plastics, and biodiesel. In addition, methanol is clean energy and intermediate source for a wide range of industrial applications (Figure 3). Therefore, converting methane into higher value-added chemicals such as methanol is necessary to develop sustainable energy sources and address global warming issues.

Methanol can be produced by selective oxidation through indirect or direct methods [5]. Industrially, indirect methods are employed for the production of methanol. The process produces intermediate syngas (CO and H_2_) through steam or dry reforming, then hydrogenated to obtain methanol. However, the syngas process is highly endothermic, requiring a large external energy supply. Therefore, this process still has many drawbacks, such as high energy consumption, complicated operation, catalyst deactivation, and reactor stability [6,7]. The direct method of methanol production is desirable to reduce production costs because the process bypasses syngas production. However, the target product is still unsatisfactory due to the low selectivity of methanol.

Zakaria and Kamarudin [8] have summarized several relevant technologies to convert methane into methanol. These include conventional catalytic processes, plasma technology, supercritical water, biological (membrane), and photocatalytic technology. However, some drawbacks are still inevitable, including (i) the relatively slow selectivity of methanol formation for conventional catalytic technology; (ii) low conversion productivity and limited by the amount of methane that can be dissolved in water for plasma technology; (iii) reactor design and high corrosion rate when treating effluents containing halogen for supercritical water processes; (iv) membrane incompatibility due to methane permeability, which can cause dissolution, swelling, or breakage of the membrane; and (v) limitations of investigation of photocatalysts and suitable photoreactors for photocatalytic technology [8].

It is desirable to synthesize methane at a lower temperature to reduce energy consumption and maintain catalyst stability. Among several technologies, photocatalytic technology seems to be a promising method for the conversion of methane conversion to methanol, as it offers an effective way to produce methanol using renewable solar energy under ambient conditions. Furthermore, photon energy that exceeds activation energy for a specific chemical reaction promotes the redox reaction. Therefore, the involvement of solar energy to facilitate photocatalytic reaction exhibits more environmentally friendly, sustainable, and renewable than fossil fuel-based energy. This technology can also reduce production costs and increase economic benefits. In addition, catalyst deactivation will be significantly reduced due to the low temperature and pressure used (under ambient conditions) in the photocatalytic process. 

Figure 4 shows the number of publications on the utilization of photocatalytic technology in general and specifically for converting methane to methanol from 2000 to 2020. The increasing number of scientific publications on photocatalytic technology for methane to methanol conversion provides clear evidence that this topic is significant. In addition, many researchers studied the effect of different structure engineering and nanomaterials on the performance of photocatalysts since their energy conversion efficiency is principally influenced by the physicochemical properties of photocatalysts and operating conditions during the photocatalytic reaction.

This paper will systematically review and discuss recent advances in photocatalytic methane oxidation to methanol (Figure 5). The scope of this review consists of the fundamentals of the photocatalytic process, particularly methane oxidation to methanol, the development of photocatalysts and their modification to achieve optimum conversion efficiency, and factors affecting the efficiency of photocatalysts to facilitate methane oxidation to methanol. Hopefully, this review will help researchers understand the fundamental photocatalytic reaction process of methane conversion to methanol in-depth and ultimately provide inspiration for developing novel composite materials based on the existing state-of-the-art photocatalysts.

## 2. Heterogeneous Photocatalyst for Photo-Oxidation Methane to Methanol 

Photocatalysis is commonly referred to as artificial photosynthesis since the photosynthetic system inspires the process. The essence of natural photosynthesis is to drive chemical reactions using optical photon energy. Upon illumination of sunlight, oxygen (O_2_) is produced from the oxidation of water (H_2_O) and carbon dioxide (CO_2_), which further transform to generate hydrocarbons with the help of chlorophylls to absorb and transfer photon energy [9]. The broad definition of photocatalysis can also be adopted from IUPAC, which can be defined as a “Change in the rate of a chemical reaction or its initiation under the action of ultraviolet, visible or infrared radiation in the presence of a substance—the photocatalyst—that absorbs light and is involved in the chemical transformation of the reaction partners.”

Currently, photocatalytic technology has been widely applied to solve problems related to air pollution [10] and environmental remediation of water contaminants [11], including dyes degradation [12], organic pollutants removal [13,14,15], heavy metals removal [16], and oil degradation in wastewater [17]. In addition, photocatalytic technology provides alternative ways to produce renewable energy, including water splitting, CO_2_ reduction, N_2_ fixation, and, in particular, CH_4_ oxidation to methanol [9].

### 2.1. Thermodynamics for Methane to Methanol Conversion

Methane has very stable and strong C-H bonds in the CH_4_ molecules, which remain challenging to convert methane into more useful hydrocarbon and hydrogen. Table 1 shows the change in Gibbs free energy for various methane reactions in 298 K (ΔG_98_). Most of the reactions have a positive ΔG_298_ indicating that the reaction is unfavorable and cannot proceed spontaneously without adding external energy as a driving force.

Methanol from methane in the industry is produced indirectly from a steam reforming process using a nickel-based catalyst followed by high-pressure catalytic conversion of the synthetic gas to methanol (Figure 6). Steam reforming is an unfavorable thermodynamic reaction and can only be proceeded at high temperature and pressure (Table 1, No. 10). Nevertheless, this process can produce H_2_ with high purity. However, there are still challenges, such as high energy consumption due to the formation of highly endothermic syngas [5], low catalyst activity, which still needs further development [19], and complex treatment to prevent heat transfer problems [19].

Methanol can also be obtained from the direct conversion of methane with water (Table 1, No. 9). Thermodynamically, converting methane into methanol using water as a reactant is an endothermic process (ΔH^0^_298_ = 77 kJ/mol). However, this process is thermodynamically unfavorable at any temperature, as reflected by the Gibbs free energy data at a temperature range of 273 K to 1500 K (Table 2). Furthermore, photocatalytic reaction offers an alternative way to facilitate these unfavorable reactions under ambient conditions. The process includes methane and water as the reactant and light as the driving force.

### 2.2. Proposed Mechanism of Photocatalytic Conversion of Methane to Methanol

The development of the conversion of methane to methanol by photocatalytic technology was initiated by Ogura & Kataoka [20]. The reaction involves water vapor at temperatures below 100 °C, methane gas, and UV irradiation at atmospheric pressure. First, methane gas continuously flowed into the water surface to obtain a mixture of water vapor and methane. Then, the mixture of water vapor and methane reacted and produced methanol as the main product. The brief mechanism of photochemical methane conversion in the research conducted by Ogura and Kataoka [20] is shown in Equations (1)–(4).
H_2_O + *hv* (λ ≥ 185 nm) → H_2_ + ½ O_2_(1)
H_2_O + *hv* (λ ≥ 185 nm) → 1/2H_2_ + •OH(2)
CH_4_ + •OH → •CH_3_ + H_2_O(3)
•CH_3_ + H_2_O → CH_3_OH + ½ H_2_(4)

The reaction begins with the photolysis of water to produce a hydroxyl radical (•OH). The hydroxyl radical then reacts with methane to produce methyl radicals (•CH_3_). Finally, the methyl radical reacts with other water molecules to produce methanol and hydrogen. Furthermore, Noceti et al. [21] combined two reactions, including water splitting and methane conversion, to produce methanol and hydrogen using La/WO_2_-based photocatalyst, as shown briefly in Equations (5)–(10). The reaction was carried out at a temperature of 94 °C, atmospheric pressure, and with visible light irradiation has succeeded in oxidizing methane with water as an oxidizing agent. As a result, the reaction produces methanol and hydrogen with a methane conversion of 4%.
La/WO_3_ + *hv* (λ ≥ 410 nm) → e^−^_CB_ + h^+^_VB_(5)
e^−^_CB_ + MV^2+^ → MV•^+^(6)
h^+^_VB_ + H_2_O → H^+^ + •OH(7)
MV•^+^ + H^+^ → ½ H_2_ + MV^2+^(8)
CH_4_ + •OH → •CH_3_ + H_2_O(9)
•CH_3_ + H_2_O → CH_3_OH + ½ H_2_(10)

However, the first obstacle in the photocatalytic conversion of methane to methanol is how to activate the highly stable methane molecule. There are two possible scenarios for activating methane using a semiconductor-based catalyst [22]. Methane can be activated directly by photogenerated holes in the VB where •OH is produced by reduction of H_2_O_2_ and oxidation of H_2_O while •O_2_^−^ produced during the reaction by H_2_O_2_ and •OH. Methanol can be obtained by: (i) reaction of methyl radical (•CH_3_) with •OH, (ii) reaction of •OCH_3_ with H_2_O, or (iii) reduction of CH_3_OOH generated by the integration of •CH_3_ and O_2_. Methane can also be activated indirectly by any active oxygenated radical species, as illustrated in Figure 7. This reaction depends on the reaction of the applied system, such as the type of semiconductor and any other additional substance added.

Gondal et al. [23] proposed a mechanism for methane activation through •OH produced by the photocatalytic process. Several studies have adopted this mechanism, as illustrated in Figure 8a [24,25]. Xie et al. [26] proposed a typical reaction of photocatalytic methanol production from methane with simultaneous production of •OH by reduction of H_2_O_2_ and •CH_3_ by oxidation of methane in the VB by the photogenerated hole. Methanol is obtained through the reaction between •CH_3_ and •OH, which probably occurs in iron species as a dopant. Zeng et al. [27] proposed a mechanism for converting methane to methanol with the addition of iron ions and H_2_O_2_ species called photocatalysis of the Fenton reaction (PCFR) (Figure 8b). The schematic mechanism for selective conversion of methane to methanol can also be achieved, as illustrated in Figure 8c. Zhou [28] proposed a methane activation mechanism by reducing O_2_ in the conduction band and oxidation of H_2_O in the valence band. Methanol will be produced by a direct combination of •CH_3_ and •OH and reduction of CH_3_OOH generated by the integration of •CH_3_ and O_2_ (Figure 8d). Moreover, the role of each active species, including e^−^, •O_2_^−^, •OH, and h^+^ was studied by utilizing K_2_Cr_2_O_7_, para-quinone, salicylic acid and Na_2_C_2_O_4_ as scavengers to trap those active species formed in the PCFR process, respectively (Figure 9). Based on these findings, e^−^ played the most dominant role in methanol production, followed by •O_2_^−^, •OH, and h^+^. 

Based on several studies, it is found that the hydroxyl radical plays a crucial role in the photocatalytic conversion of methane to methanol. This species can be obtained either by photooxidation of H_2_O or photoreduction of H_2_O_2_. In general, H_2_O_2_ is added to facilitate the formation of •OH and, thus, enhance photocatalytic efficiency. For example, Noceti et al. [21] reported that adding H_2_O_2_ to the reaction carried out at atmospheric pressure, and temperature of 94 °C with irradiation from a mercury lamp (46% visible light) increased methane conversion from 4% to 10%, accompanied by increased CO_2_ formation. Zeng et al. [27] also reported excellent results in reactions using TiO_2_ photocatalyst and addition of H_2_O_2_ under visible light irradiation, with methanol yields up to 850 mol g^−1^ h^−1^ and a selectivity of 83%. 

On the other hand, a study conducted by Gondal et al. [29] showed the opposite effect, where the addition of H_2_O_2_ exhibited a lower yield of methanol than the reaction without H_2_O_2_ addition using WO_3_ photocatalyst under visible laser irradiation and ambient conditions. These results are believed to be due to the usage of a visible laser lamp providing higher photon flux density monochromatic light than the conventional lamp so that more •OH was formed. These •OH radicals are also a major source of O_2_ that facilitates further oxidation of methanol. These results agree with Villa et al. [30], which carried out a reaction using WO_3_ photocatalyst at atmospheric pressure, the temperature of 55 °C, and under UVC-Vis irradiation. It was found that the formation of ethane accompanied the methanol production due to the higher amount of •OH activated methane to generate more •CH_3_, as shown in Equation (11).
•CH_3_+ •CH_3_ → C_2_H_6_(11)

Based on these findings, Villa et al. [24] proposed a strategy to incorporate fluorine on the WO_3_ photocatalyst surface to minimize the interaction between the catalyst and the reactants. The schematic reaction mechanism on the surface of the WO_3_ before and after the incorporation of fluorine is shown in Figure 10. It was found that •OH groups on the catalyst surface are mainly responsible for enhancing the performance of WO_3_ in the selective oxidation of methane to methanol. At the same time, an enormous amount of free OH radicals favor the formation of ethane. Therefore, adding a hydroxyl radical generator, i.e., H_2_O_2_, can increase methane conversion [31]. However, such an appropriate amount of H_2_O_2_ is essential for controlling the selectivity of methanol [22,23,27].

Methanol, the primary desired main product of methane oxidation, is more reactive than methane itself. Thus, methanol can be easily oxidized further when some of the following conditions occur, such as:The formation of superoxide radicals (•O_2_^−^) produced via the reduction of O_2_ in CB of semiconductor [23], shown in Equations (12)–(14).
O_2_ + e^−^ → O_2_•^−^(12)
CH_3_OH + O_2_•^−^ → CO_2_ + H_2_O + e^−^(13)
CH_4_ + 2O•^−^ → CO_2_ + H_2_O + 2e^−^(14)

2.A sufficient amount of methanol generation either competes with water to further react with a photogenerated hole in VB or direct interaction with •OH to produce side products such as formaldehyde [32], as shown in Equations (15)–(17)

CH_3_OH + h^+^ → CH_3_OH^+^ → H_2_CO + H• + H^+^(15)

H_2_CO + h^+^ → CO + H• + H^+^(16)

CH_3_OH + •OH → CH_2_O + H• + H_2_O(17)

According to the abovementioned issues, it is essential to control the oxidative power of the system to achieve optimum methanol selectivity. For instance, using a moderate photocatalyst lowers the potential reduction of H_2_O_2_ [26] and prevents O_2_ reduction to form •O_2_^−^ by adding an electron scavengers agent [29]. In addition, the yield of methanol can be enhanced by continuous removal of the methanol after its formation to prevent further oxidation of methanol [32].

## 3. Selection of Materials for Photocatalytic Methane to Methanol Conversion

Heterogeneous photocatalysts have been widely studied and applied to photocatalytic processes. This catalyst only requires energy from photons to drive the reaction under ambient conditions, unlike conventional catalysts that require heat or high-temperature conditions. Upon illumination of light with sufficient energy (higher or equal to band gap energy of semiconductor-based photocatalysts), the photon energy from the light is absorbed by the semiconductor and initiates the excitation of electron (e^−^) from VB to CB, leaving a hole (h^+^) in VB. Photogenerated electron-hole or charged pairs are responsible for promoting redox reactions on the photocatalyst surface.

The ideal semiconductor photocatalyst must be chemically and biologically inert, photo-catalytically stable, easy to manufacture and use, can be activated by low energy of light, i.e., sunlight, catalyze reactions efficiently, inexpensive, and environmentally friendly [33]. The crucial semiconductor properties required to establish a successful chemical photoreaction include band gap energy, energy band position of the semiconductor, and the appropriate redox potential of the adsorbate to produce the desired product. Thus, it is fundamental to identify the Highest Occupied Molecular Orbital (HOMO) and Lowest Unoccupied Molecular Orbital (LUMO) of the semiconductors involved in the charge transfer during the reaction. HOMO coincides with the top of the VB, where all electronic levels are occupied, while LUMO coincides with the bottom of CB, where all electronic levels are empty [34]. 

Based on the appropriate redox potential, several semiconductors have the potential to carry out the photocatalytic conversion of methane to methanol, such as TiO_2_, WO_3_, BiVO_4_, ZnO, Fe_2_O_3_, NiO, and CuO (Figure 11). Those semiconductors have an absolute energy position of the VB below the potential oxidation of H_2_O/•OH to allow hole transfer to the reactant to carry out the oxidation reaction and also an absolute energy position of CB above the potential reduction of O_2_/*•*O_2_^−^ and H_2_O_2_/•OH to allow the donation of the excited electron to carry out the reduction reaction. 

Heterogeneous photocatalysts can be obtained from pure/basic metal oxide or modified metal oxide. The intrinsic properties of metal oxide, such as a range of light absorption, electronic band structure, specific surface area, particle size, and morphology, are considered for its application as a photocatalyst. Generally, getting a pure metal oxide with high applicability is challenging to fulfill all photocatalytic reaction prerequisites. Therefore, it is necessary to modify the semiconductor to adjust and improve their properties toward defined requirements for their intended use. Common metal oxides and their modification, including their properties, photocatalytic activity and their drawbacks toward methanol production, will be discussed in the below section.

### 3.1. Pure Semiconductor

Due to their unique electronic properties, semiconductors have been widely used as photocatalyst material in photocatalytic reactions. Therefore, the performance of different pure semiconductors is studied and compared in the same reaction condition or with impure semiconductors. Among various semiconductors, tungsten, titanium, bismuth, zinc, and iron-based semiconductor are the most frequently studied for photocatalytic oxidation of methane to methanol since they show promising activity. Therefore, this section will review each pure semiconductor’s performance toward methane photo-oxidation to methanol. 

Tungsten-based semiconductors are commonly found as photocatalysts in the form of WO_3_. It is one of the most common semiconductors used as a photocatalyst for methane oxidation to methanol due to its unique chemical, functional, and physical properties, low cost, small band gap energy (2.4–2.8 eV), and stable physicochemical properties [36]. In addition, it has moderate oxidizing power, preferable for methanol production from methane oxidation [37]. WO_3_ was also demonstrated to effectively absorb visible light, which is the most energy in sunlight. However, pure WO_3_ is still difficult to apply since it is confronted with the rapid recombination of photogenerated electron-hole pairs [36]. Thus, modification of WO_3_ to prevent the recombination losses of charge carriers is necessary and crucial. 

Titanium dioxide (TiO_2_) is the most widely used semiconductor for photocatalyst due to its wide band gap, lower price, high photocatalytic activity, considerable photostability, non-toxicity, ample availability, insolubility in aqueous solution, and chemical and biological inertness [38]. Currently, TiO_2_ materials are used for many applications, including water splitting [39], water treatment [40], air pollution treatment [10], solar cells [41], and many other applications (Figure 12).

The crystal structure of TiO_2_ can be classified into three phases: anatase, rutile, and brookite (Figure 13), where each has different characteristics and band gaps. Anatase is stable at low temperature and has a band gap of 3.2 eV, while rutile is stable at high temperature and has a band gap of 3.0 eV. Meanwhile, brookite is usually found in minerals with orthorhombic crystal structure and has a band gap of 3.25 eV [38,43,44]. Anatase has the highest photocatalytic activity among other TiO_2_ crystal phases because it has the lightest average effective mass of photogenerated electrons, a higher surface adsorption capacity to hydroxyl groups, and a lower charge carrier recombination rate than rutile and photoexcited charge carriers are more easily to migrate and transfer to the surface [45,46]. In addition, anatase TiO_2_ has a wide band gap which puts this semiconductor as a UV light-absorbing semiconductor. Therefore, it displays a high photocatalytic activity when irradiated by UV light. However, it remains a challenge that needs to resolve for semiconductor TiO_2_ because it is desired for a photocatalytic process to absorb visible light wavelength.

The bismuth-based semiconductor is a visible-light responsive photocatalyst that provides an excellent opportunity to convert sunlight into chemical energy. Every bismuth-based material usually possesses narrow band gaps less than 3.0 eV due to their electronic structure in the valence band, which consists of oxygen 2p and bismuth 6s [47]. In the photocatalytic process, bismuth-based material can be prepared as Bi_2_O_3_, Bi_2_MO_6_ (M = Cr, Mo, and W), BiVO_4_, BiOX (X = Cl, Br and I), BiPO_4_, (BiO)_2_CO_3_, and pentavalent bismuthates [44]. Bismuth-based semiconductors are widely used as photocatalysts for many applications such as water splitting [48], pollutant degradation [14], CO_2_ reduction/N_2_ fixation [49], and organic synthesis (Figure 14).

Zinc-based photocatalyst is commonly used as a photocatalyst in the form of zinc oxide (ZnO). ZnO has a broad direct band gap (3.37 eV) which corresponds to ultraviolet absorption and large excitation binding energy (60 meV) [51,52]. As a result, it shows excellent electrical, mechanical, and optical properties, similar to TiO_2_ [53]. Besides that, it possesses a lower production cost of photocatalyst [54] and higher absorption efficiency across a more significant fraction of the solar spectrum than TiO_2_ [51]. The ZnO semiconductor is generally studied in photocatalytic reactions for dye degradation coupling with various kinds of dopants, such as transition metal [55], alkaline earth [56], a nonmetal [57], zeolite [58], rare metal-earth [59], and other metals [60]. However, the study of ZnO for photocatalyst toward photocatalytic oxidation methane to methanol is still limited compared with WO_3_ and TiO_2_. Moreover, the ZnO photocatalyst application encounters some constraints, such as its limited light-harvesting ability, photo-corrosion severe problem, and rapid recombination of electron-hole pairs [61].

Iron oxides exist in many forms in nature, such as magnetite (Fe_3_O_4_), hematite (α-Fe_2_O_3_), and maghemite (γ-Fe_2_O_3_) [11]. It has low cost, high stability, and compatibility with the value of the direct and indirect band gap nanomaterials ranging from 1.95 to 2.35 eV and 1.38 to 2.09 eV, respectively [62]. This semiconductor has been widely applied for wastewater treatment [63,64]. However, the iron-based photocatalyst is hardly ever found as the primary catalyst used in the photocatalytic oxidation of methane to methanol. Commonly, it is used as a dopant, cocatalyst, and electron scavenger in the reaction system and noticeably shows a remarkable improvement in the reaction.

The performance of those metal oxide semiconductors has been studied in the form of their pure/bare metal oxide in the photocatalytic oxidation of methane to methanol. For example, Gondal [23] studied the performance of several pure semiconductors (WO_3_, TiO_2_, and NiO) in the same reaction condition. Photocatalyst WO_3_ exhibited the highest methane conversion with the conversion of WO_3_, TiO_2_, and NiO, 29%, 21%, and 20%, respectively. On the other hand, Xie [26] studied the pure anatase of TiO_2_ compared with iron-doped-TiO_2_. The reaction was conducted over 3 h of irradiation and showed a methane conversion rate of 10.9%, but lower methanol selectivity of 36% compared with iron-doped-TiO_2_. 

Lopez [65] evaluated several synthesized bismuth-based materials in the selective photooxidation of methane. The reaction was performed in a photochemical reactor equipped with an immersion medium-pressure mercury lamp (450 W) at 55 °C for 120 min of irradiation, with CH_3_OH, CO_2_, and C_2_H_6_ as the main products. BiVO_4_ showed better methanol selectivity and lower CO_2_ and C_2_H_6_ levels (Figure 15). Moreover, it also showed a minimal change of color in contrast with BiWO_6_, which showed a shift in color suspension at the end of the test.

A study by Zeng [27] reported using Fe_2_O_3_ as a photocatalyst in the conversion of methane to methanol and showed a lower methane conversion and methanol selectivity than TiO_2_ photocatalyst. Song [66] studied pure ZnO and TiO_2_ catalyst compared with the doped photocatalyst. The reaction was conducted in a 230 mL batch- reactor equipped with a quartz window, 2 MPa pressurized methane, and under a 300 W Xenon lamp. Methanol was not produced in both photocatalysts, while CO and CO_2_ were detected with higher production in TiO_2_. 

The abovementioned metal-oxide is the most common material used as a photocatalyst for photocatalytic methane to methanol conversion. However, it is also worth mentioning metal-organic frameworks (MOFs) as photocatalysts, although the application of MOFs is still limited compared to metal oxide. MOFs consist of metal ions or clusters coordinated to organic ligands or linkers. It has emerged as a photocatalyst owing to its structural characteristic of large surface area, well-ordered porous structure and tunable organic linkers/metal clusters. Those properties endow promising photophysical/chemical properties to facilitate adsorption of CH_4_ onto the catalyst surface, charge separation and reactant activation [67,68]. A study about photo-oxidation of methane to methanol using MOFs as photocatalysts is reported by An [69] over mono-iron hydroxyl sites immobilized within a metal-organic framework, PMOF-RuFe(OH). A remarkable selectivity was shown that methane was converted to methanol with 100% selectivity and a yield of 8.81 ± 0.34 mmol g_cat_^−^h^−^. The mono-Fe^III^ species acts as a binding and activation site for CH_4_. However, MOFs have some important issues needed for further improvements, such as poor electronic conductivity, which hampers the charge transfer from organic linker to metal cluster and poor stability during the photocatalytic reaction [70].

### 3.2. Modified Semiconductor

As discussed above, pure semiconductors generally cannot exhibit the excellent properties required to support photocatalytic reactions. In addition, critical properties, such as narrow absorption spectra, low photon quantum efficiency, and easy recombination of electrons and holes, become the main drawbacks of neat/pure semiconductors. Thus, several modifications of pure semiconductors have been studied to improve pure semiconductor properties, including metal/non-metal doping, heterojunction, and crystal facet rearrangement.

#### 3.2.1. Metal/Non-Metal Doping

Elemental doping is a common modification of photocatalysts to optimize its properties by introducing metal or non-metal into the semiconductor. Additional metal plays a role in introducing a new energy level into band structure to extend the light absorption of photocatalysts and help trap electrons or holes, leading to reduced recombination of photogenerated charge carriers [71,72]. Thus, incorporated dopant will enhance the light absorption, regulate band position, and charge carrier process of semiconductors [72]. This section discusses the metal/non-metal doping for each common metal-oxide photocatalyst used in the photo-oxidation of methane to methanol. 

Several efforts have been performed to examine the effect of dopants on photocatalyst performance in absorbing light irradiation, visible light in particular. The former study by Taylor [73] used four dopants, including copper, lanthanum, platinum, and a mixture of copper and lanthanum in a WO_3_-based photocatalyst. The structure of the doped WO_3_ photocatalyst showed to be more crystalline with a larger crystal and smoother edge. Prior to the study, a reaction without photocatalyst was performed at a temperature between 65 and 95 °C. Conversion of methane and methanol production decreased as temperature decreased, and no photochemical product was observed below 70 °C. Reaction performed with doped semiconductor photocatalyst at temperature 94 °C has successfully converted methane and water to methanol, hydrogen, and acetic acid. 

Taylor and Noceti [74] performed the same doped WO_3_ photocatalyst, using an additional filter to remove UV components from the lamp. A positive result appeared on the lanthanum doped catalyst that methanol production is higher than non-catalytic reaction. However, platinum and lanthanum/copper exhibited similar results in a non-catalytic reaction. The detrimental effect appeared on copper doped in tungsten oxide inhibited methanol production. In comparing unfiltered and filtered lamp irradiation, experimental platinum doped tungsten oxide showed little difference. This explained that the photocatalyst was operating in visible light the presence of UV light was negligible.

Negishi [25] attempted to activate m-WO_3_ and m-TiO_2_ by loading the ultrafine metal clusters as cocatalyst on the photocatalysts, including Ag, Ni, and Co. Photocatalyst was synthesized using the hard template method, and metal was loaded using the solid and liquid phase methods. Loading metal clusters have proven to improve the yield of methanol compared to single m-WO_3_ (Figure 16). It happened because loading metal can promote charge separation effectively, and, hence, holes are consumed effectively [75]. The highest methanol yield was shown on the Co_n_-m-WO_3_ because the relationship between the valence band of WO_3_ and Co orbital and the relationship between the redox potential of •OH and Co was effective in accelerating the reaction [25].

Atomic-scale gold dispersed in WO_3_ was fabricated by Yi [76] via a photochemical reduction route. Although gold (Au) is an expensive and scarce noble metal, the utilization of this metal as a cocatalyst is still studied because of its high electronegativity. This catalyst showed a better photocatalytic performance than a gold nanoparticle. Further, it is confirmed that the atomic-scale gold on WO_3_ generates the typically active species, such as hydroxyl radical (•OH), hydroperoxyl radical (•OOH), and methyl radical (•CH_3_), which play an important role in the photocatalytic reaction of methane to methanol.

Despite its outstanding properties, photocatalyst TiO_2_ has several drawbacks that limit its application, such as the wide band gap, high recombination rate of electron-hole pairs, and the weak separation efficiency of photocarriers [71]. However, modifying its wide band gap and electronic structure of TiO_2_, which bring on poorly visible light-harvesting properties, can reduce massive recombination and improve the interface and surface characteristics [77]. 

Loading several co-catalysts in the m-TiO_2_ by Negishi [25] noticeably showed better photocatalytic activity than pure m-TiO_2_. The highest co-catalytic effect was exhibited by loading cocatalyst Ni in m-TiO_2_ with methanol production was around 0.14 μmol·h^−1^ compared with pure m-TiO_2_, which only generated methanol around 0.02 μmol·h^−1^ (Figure 17). However, TiO_2_ showed a lower methanol generation than WO_3_ for both pure and loaded with a co-catalyst. From the experimental results, it can be inferred that the loading of such appropriate co-catalysts effectively improves the activity of photocatalysts [25].

Recently, Wu [78] reported a combination of 2D Titania sheets and atomic-scale Pd species for selective oxidation of methane to methanol. This catalyst is synthesized in 2 steps by solvothermal method to fabricate crystalline 2DT sheets, followed by photochemical reduction to deposit Pd on 2DT. Pd/2DT catalyst exhibited good selectivity and stability in solar light irradiation. Stability is obtained from 2DT that avoids photo-corrosion, and OVs formed in situ in the 2DT material stabilize atomic Pd and suppress the aggregation during methane oxidation. An impressive activity of the catalyst is obtained by the role of Pd in promoting the separation of photo-induced electron-holes and providing active sites for reduction of H_2_O_2_ toward the •OH radical.

Some metals have been studied as co-catalysts to modify zinc oxide-based photocatalysts [24,56]. For example, Song et al. [66] studied the selective oxidation of CH_4_ to liquid oxygenates using ZnO and some cocatalysts (Pt, Pd, Au, or Ag) at room temperature in water with the addition of O_2_. Up to 125 μ molh^−1^ of liquid oxygenates, including methanol and formaldehyde, are produced with selectivity higher than 95% over 0.1 wt % Au/ZnO. Efficient and controlled activation of O_2_ over cocatalysts has mild reactive oxygen species, •OOH radicals. The •OOH radicals, with the assistance of protonation of water, will enhance the selective liquid oxygenate generation and avoid overoxidation of CH_4_. 

The bismuth-based semiconductor is a visible-light responsive photocatalyst that already has narrow band gap energy. Thus, doping may not be considered to be controlled for bismuth-based photocatalysts. To the best of our knowledge, there has been no study report for doping modification in a bismuth-based photocatalyst. However, from several experiments, doping element metal in the photocatalyst can improve the intrinsic properties of the pure semiconductor. Adding an appropriate co-catalyst positively affects photocatalytic activity indicated by enhancing methane conversion, methanol production, or selectivity. 

From several studies discussed, adding an appropriate dopant to the photocatalyst has been proven to enhance the photocatalytic activity of the photocatalyst, which is indicated by the improvement of methane conversion, methanol yield, or methanol selectivity compared to pure catalyst. Some metals from base metal (copper, iron, nickel, lead, and zinc) and noble metal (gold, silver, platinum, and palladium) have been extensively studied. Noble metal has been used widely as a catalyst due to its unique intrinsic properties, irreplaceable catalytic activities, good stability, and oxidation resistance. However, the application is still limited to the high cost and the availability. Ultimately, loading dopant will improve the properties of semiconductors, such as enhancing its absorption in visible light, effectively promoting charge separation, and driving to produce preferable species in the reaction. 

#### 3.2.2. Heterojunction

Reducing and preventing recombination charge pair carriers have been a concern in the photocatalytic reaction. The recombination process can be illustrated as a person jumping from the surface earth (Figure 18). The force of gravity will cause the person to fall back to the earth’s surface. However, when a bench is available, the person can land and not fall on the ground. This is the same as excited electrons from the valence band. The Coulomb force will cause electrons to return, but electrons will fall in the new conduction band when a conduction band is available from another semiconductor. 

Heterojunction modification has been extensively studied as an effort to improve the properties of the semiconductor. It creates an interface between two semiconductors with different band structures to generate a band alignment. There are several types of heterojunctions, including conventional (type I, II, III), type p-n, surface, and direct Z-scheme heterojunction, as shown in Figure 19 [79]. Type II is the most effective heterojunction among conventional heterojunctions because it allows effective electron-hole separation, wide light-absorption range, and fast mass transfer [79]. Type p-n has a more efficient and faster electron-hole separation than type II heterojunction because it creates an internal electric field from the diffusion of the electron from n-type to p-type semiconductor [80]. Different crystal facets on semiconductors have other band structures which can also be used as heterojunctions, similar to that of type II heterojunctions, named surface heterojunctions. One highlight of this heterojunction type is the low cost because it is fabricated from one semiconductor.

Some examples of type II heterojunctions are discussed here. Hameed [32] synthesized WO_3_ photocatalyst impregnated with Ag^+^ ion in several proportions (0.1%, 1%, 5%, and 10%) by dissolving AgNO_3_ (99.9%) in deionized water and mixed with the appropriate quantity of WO_3_. FE-SEM confirmed the Ag^+^ ion present as Ag_2_O. XPS showed that the surface of Ag_2_O particles resides at surface WO_3_ without affecting the morphology base WO_3_ (Figure 20). The more loading Ag^+^ ion into WO_3_, the longer wavelength is shifted on the absorption spectra characterization. It means that loading Ag^+^ can enhance absorption in the visible region. However, in the 10% loading Ag as the highest loading in this experiment, metallic Ag^0^ was observed at the inner surface of the photocatalytic reactor, which may be due to the depletion of silver content on the surface, as shown in Figure 20. The photocatalytic charge transfer process using Ag_2_O is shown in Equations (18) and (19).
Ag_2_O + hv → 2Ag^+^ (sol) + ½ O_2_^−^ (surface)(18)
Ag^+^ (+0.8 V) + e_cb_^−^ (+0.7 V) → Ag^0^(19)

Combining Ag_2_O (+1.39 V) and WO_3_ (+3.1 V) with different band edge positions will generate heterojunction modification that can enhance the lifetime of excited states. The mechanism of an extended lifetime by heterojunction is illustrated in Figure 21. The interface between metal oxide allows the migration of hole (h^+^) generated by photon-induced electronic excitation from the valence band of WO_3_ to the valence band of Ag_2_O. The electrons (e^−^) from the conduction band Ag_2_O will then be transferred to the conduction band of WO_3_. Hydroxyl radical formation can be enhanced by suppressing the charge carrier recombination. The highest methanol production from this process was gained at 5% metal loading in WO_3_. The metal loading higher than 5% affected the proton that it tends to be consumed in dissociation of Ag_2_O than being productive as suppressing recombination species. 

Xie [26] reported metal oxide species anchored on TiO_2_ for photocatalytic conversion of methane to methanol at ambient conditions (1 bar and around room temperature) in an aqueous phase system with H_2_O_2_ as oxidant. Several metal oxides are Au, PdO_x_ PtO_x_, Cu_2_O, and FeO_x_. Iron oxide species showed the best photocatalytic activity, not only high methane conversion (15%) but also high methanol production (>1000 µmol g_cat_^−1^) obtained (Figure 22), which may be due to a match in Fermi levels, the high dispersity, and small size of the nanoclusters [26].

Besides Bi_2_WO_6_ and BiVO_4_, Lopez [65] also evaluated Bi_2_WO_6_ coupled with TiO_2_. Unfortunately, Bi_2_WO_6_/TiO_2_ did not show a better photocatalytic activity than Bi_2_WO_6_ and BiVO_4_, which dominated overoxidation CH_4_ to CO_2_. However, Bi_2_WO_6_/TiO_2_ photocatalyst showed higher photocatalytic activity than sole water [65].

Heterojunction modification can evidently enhance the photocatalytic activity of photocatalysts. Loading another semiconductor will give a new conduction band that can prevent recombination and prolongs the charge carriers’ lifetime. However, a suitable combination of photocatalysts and the optimum amount of cocatalysts should be considered to get the maximum result.

#### 3.2.3. Structure Modification

Structure modification is another modification that can be applied to improve the properties of the photocatalyst. Structure modification discussed here will be referred to as modification resulting from synthesis, such as crystal structure and formation of mesopore photocatalyst. In addition, the mechanism of the photocatalytic reaction of methane oxidation is also highlighted according to modification on the photocatalyst surface. 

Li [82] has successfully synthesized high crystalline WO_3_ with organized mesopore using PI-b-PEO as a structure-directing agent (Figure 23). Applying mesopores structure to WO_3_ photocatalyst has proven to eliminate diffusion limitation so that photocatalytic performance can be enhanced. It might have happened because the high surface area can facilitate access and adsorbed reactants on the surface [82].

High crystalline with ordered mesopores WO_3_ was studied by Villa [37] for photocatalytic conversion of methane to methanol. The synthesized catalyst characterization found WO_3_ as a monoclinic structure, consisting of a well-ordered structure with irregularly shaped pores and a surface area of 151 m^2^ g^−1^. 

Adding H_2_O_2_ to the reaction system contained the ordered mesopores WO_3_ exhibited a higher methanol yield than single mesopore WO_3_. It proved that with a larger surface area of photocatalyst, the more hydroxyl radical could be adsorbed in the catalyst and consequently will lead to the higher formation of CH_3_OH [37].

This study substantiated the study of the role of WO_3_ surface hydroxyl groups for the partial photocatalytic oxidation of methane to methanol by Villa [24]. Furthermore, this research concluded that OH radical groups on the catalyst surface are mainly responsible for enhancing the performance of WO_3_ in the selective oxidation of methane to methanol [24].

#### 3.2.4. Crystal Facet Re-Arrangements

Photocatalytic activity of semiconductors cannot disregard in the surface atomic structure tuning. The modifying surface structure of the semiconductor is intended to adjust the exposed facets. Crystal facet engineering could affect the electronic band structure, charge transfer, and separation by tuning the surface free energy, formation of hetero/homojunction, and effective mass of hole/electrons [83]. Since the modification is applied on the surface of semiconductors, crystal facet re-arrangements are considered surface heterojunction. 

Anatase TiO_2_ has some fundamental exposed crystal facet including {001}, {100}, and {101} with different average surface energy valued 0.9 J/m^2^, 0.53 J/m^2^ and 0.44 J/m^2^ [84], respectively. Recently, Feng [85] explored {001} and {101}- dominated TiO_2_ with Ag cocatalysts for photooxidation of methane by oxidant O_2_ in an aqueous solution containing H_2_O. The {001}-dominated Ag/TiO_2_ showed better result than {101}-dominated TiO_2_ with methanol yield 4.8 mmol g^−1^ h^−1^ and selectivity ~80% (Figure 24). This result was believed to be from the role of {001} in avoiding the formation of •CH_3_ and •OH, which contribute to methanol overoxidation [85].

Zhu [86] reported that controlling the crystal facet of BiVO_4_ microcrystal enhanced methane to methanol conversion into a bipyramid, a thick platelet, and a thin platelet shape prepared by hydrothermal synthesis (Figure 25). Referring to the XRD and Raman spectroscopy characterization, the synthesized BiVO_4_ was found as a monoclinic phase. The surface area measured using the BET method for bipyramid, thick platelet, and thin platelet were determined to be 3.2 m^2^ g^−1^, 3.6 m^2^ g^−1,^ and 3.6 m^2^ g^−1^, respectively. 

The reaction was conducted at a temperature of 65 °C using a Xenon arc lamp (350 W), and the products CH_3_OH, CO_2_, and H_2_ were measured in the reaction. The highest activity was achieved for the BiVO_4_ bipyramids with 151.7 μmol h^−1^ g^−1^ at 2 h reaction time. Similar selectivity emerged on the bipyramid and thick platelet with 85.0% and 85.7%, respectively. Thin platelets showed high activity for water oxidation [87] otherwise, thin platelets produced a large amount of CO_2_ rather than CH_3_OH as the main product in this experiment. 

From such a result, it was believed that the bipyramid possesses an efficient extraction of holes on the entire surface and intermediate surface reactivity, leading to high methanol selectivity [86]. On the other hand, thinner BiVO_4_ has a strong oxidation ability [87].

To the best of our knowledge, no study has reported on the crystal facet re-arrangement of WO_3_ for photocatalytic conversion of methane to methanol. However, crystal facet engineering has become a potential modification in a photocatalyst. Photocatalysts enclosed with preferred index facets in high percentages could significantly improve their photocatalytic activity. Furthermore, it was proven by some studies conducted for photocatalytic oxidation of methane to methanol that the selectivity of methanol can be enhanced. 

#### 3.2.5. Electron Scavenger

Some photocatalytic for methane conversion to methanol reported so far employ an electron scavenger. The addition of electron scavengers has been shown to affect the photocatalytic activity and selectivity of methanol significantly. For example, Villa [37] reported using several electron scavengers, including Fe^3+^, Cu^2+,^ and Ag^+^ (Figure 26). Significant improvement in methanol yield was shown by a particular additional amount of Fe^3+^ and Cu^2+^ by a factor of 2.5 and 1.7, respectively. Contrary, the Ag^+^ showed a detrimental effect on methanol generation compared to single WO_3_. Moreover, Ag deposited emerged on the surface of the catalyst during the reaction.

Zeng [27] studied the synergetic effect of Fe^2+^ and H_2_O_2_ called the Fenton reaction to the photocatalytic reaction for several photocatalysts, including Fe_2_O_3_ and TiO_2_ NiO, CuO, ZnO, and WO_3_. Photocatalyst TiO_2_ showed the highest yield for methanol, while photocatalyst WO_3_ showed the highest yield for formic acid (Figure 27). This experimental result exhibited that adding appropriate Fenton reagents will contribute to methane conversion and methanol selectivity [27].

Electron scavenger can leverage the photocatalytic reaction because this species will capture the electron so that recombination of charge carrier (e^−^ and h^+^) can be avoided and methanol yield will enhance simultaneously [88]. Furthermore, adding Fe^3+^ ions in the reaction can inhibit the generation of •O_2_^−^ by capturing electrons in the conduction band to form Fe^2+^ [29]. The electron scavenger is also an appealing modification for the photocatalytic reaction because this method is relatively simple and cost-effective [23,33].

Zeng [27] has successfully achieved a highly selective conversion of methane to methanol through photocatalysis-Fenton (Fe^2+^ and H_2_O_2_) reaction with solar energy at room temperature. The Fenton reaction was carried out under solar light irradiation. Methane with a pressure of 3 MPa was injected into the reactor, and the suspension temperature was maintained at 30 °C. As a result, excellent photocatalytic activity was achieved using TiO_2_ photocatalyst with methanol yield and selectivity of 471 μmol g^−1^ h^−1^ (Figure 27) and 83% (Figure 28), respectively. 

It was believed to have occurred due to the band structure of TiO_2_, which exactly matches well with the redox potentials of H_2_O_2_/•OH, H_2_O/•OH, and O_2_/•O_2_^−^ and the Fenton reaction promoted the closed cycle (Fe^3+^ + e → Fe^2+^) and iron restoration (Figure 29). 

In evaluating bismuth-based synthesis for selective photoactivation of methane conducted by Lopez [65], two-electron scavengers O_2_ and Fe^3+^ were introduced to the photocatalyst Bi_2_WO_6_ system. Adding electron scavengers for both O_2_ and Fe^3+^ showed a similar behavior that no change of color observed at the end of the test, indicating that O_2_ can serve as electron trapping and Fe^3+^ can inhibit self-reduction of Bi_2_WO_6_ [65]. However, adding Fe^3+^ aggravated the selectivity of methanol since CO_2_ dominated CH4 overoxidation, which can be due to Fe^3+^ improved electron transfer, enhanced charge separation, and produced higher hydroxyl radical [65].

Besides the electron, it is also essential to trap •OH in the photocatalytic system, as Lopez [88] reported using nitrite ion. Hydroxyl radical is crucial to produce methyl radical, but excessive •OH could lead to aggravation of methanol selectivity. Thus, additional nitrite ions in the photocatalytic system containing BiVO_4_ could improve the selectivity of methanol than pure only BiVO_4_.

The addition of electron scavengers offered a more accessible and cost-effective way to achieve the same benefit as the co-catalyst [27]. Iron oxide species are the most active electron scavenger for photocatalytic methane oxidation to methanol compared with other metals [27,29,37]. Iron species have some roles in this process, such as an electron acceptor to prevent recombination of charge pairs and inhibit the generation of •O_2_^−^ to over-oxidized methanol.

Table 3 shows recent progress on development of neat and modified photocatalysts for photocatalytic oxidation of methane to methanol with various synthesis method; operating condition including photocatalyst loading, reactant, light source, operating pressure and temperature, reactor type; in conjunction with photocatalytic performance including yield, conversion, and selectivity. 

## 4. Factors Affecting Photocatalytic Methane Oxidation to Methanol

### 4.1. Physiochemical Characteristics of Photocatalyst

#### 4.1.1. Band-Gap

The semiconductor has a unique electronic property that will allow the electron to exit upon given appropriate energy. Therefore, for photocatalytic application, energy as a driving force is provided in the form of light irradiation. This phenomenon can occur due to the presence of a band gap in the semiconductor. The band gap is a gap between the valence and conduction bands where no electrons can reside or is also called a forbidden gap. 

As explained before, upon illumination of appropriate light with the same or higher energy than the band gap, electrons will excite from the valence band to the conduction band. Consequently, generating electrons (e^−^) in the conduction. The band gap value can be calculated by applying the Kubelka–Munk function, as shown in Equation (20) [90].
(20)$$\alpha \left(hv\right)\approx {\left(hv-E_{g}\right)}^{n}\;\;$$

*E_g_* = Band gap energy (eV)

*h* = Planck’s constant

α = Extinction coefficient

*v* = Light frequency (s^−1^)

*n* = 2 (for *indirect allowed transition*)

*n* = 3 (for *indirect forbidden transition*)

*n* = ½ (for *direct allowed transition*)

*n* = 3/2 (for *direct forbidden transition*)

Another method for band gap determination was proposed by Zanatta [91] by fitting a sigmoid (Boltzmann) function to the semiconductor’s corresponding optical. This method is claimed to be more accurate and straightforward than the traditional approaches. 

A semiconductor with a wide band gap absorbs the UV light spectrum, while a low band gap absorbs visible light. However, some semiconductors can only effectively absorb light under UV light. Therefore, an effort has been focused on decreasing the band gap of the photocatalyst. Thus, visible light can be used instead of UV light by doping the semiconductor with another metal or non-metal. 

The critical consideration in selecting the photocatalyst for the photocatalytic reaction is the position of the photocatalyst band gap that can produce the intended species for reaction. Methanol production from methane photooxidation can be produced either by species O_2_ or •OH produced from H_2_O and H_2_O_2_. Thus, among several photocatalysts, TiO_2_ and WO_3_ have been commonly used in this process because they have the most suitable band gap position to produce the required species for reaction (Figure 30). However, O_2_ is a strong oxidant that could lead to over oxidation of methanol produced in the system. The band gap position of WO_3_ is unsuitable for degrading methanol via oxidation by O_2_. It could be why the WO_3_ is the most frequent photocatalyst studied in methanol production from methane photooxidation. 

#### 4.1.2. Morphology and Specific Surface Area

The morphology of photocatalysts is one factor that strongly affects photocatalytic performance [92]. Reactants should overcome the diffusion limitation upon reaching the surface of the photocatalyst, and sometimes it becomes the limiting step during the photocatalytic process. Controlled pore size and high surface area photocatalyst can facilitate the reactant to access the surface easily. Hence, photocatalytic performance can be maximized [93]. Porous materials are classified into several kinds by their size, including microporous (<2 nm), mesoporous (2–50 nm), and macroporous (>50 nm) [94]. The meso- and macropore sizes are desirable because these pore sizes are considered to facilitate the diffusion of the molecule [95].

Mesopore photocatalyst in ordered channels is highly photocatalytic because it can facilitate fast intraparticle molecular transfer to decrease charge recombination [37]. In addition, pore connectivity in the porous material leads to the large surface of synthesized ordered m-WO_3_. The hierarchically porous materials and ordered nanostructure have overcome mass transport limitations [82,96]. In terms of pore arrangement, ordered mesopores show higher activity than disorder mesoporous [92] because the ordered mesoporous can enhance light harvesting from a deeper light penetration and light scattering in the pore channels facilitate diffusion and adsorption of the reactant [97]. 

The crystallinity of photocatalysts is also essential to improving the properties and activity of photocatalysts. Improving the crystallinity of photocatalysts can prevent charge recombination from occurring [98]. Several studies have proven that high crystallinity can improve photocatalytic activity, leading to a higher yield of product [82,99,100], Thus, the synergistic effect of high crystallinity and ordered mesoporous channels could significantly reduce the electron-hole pair recombination to obtain a higher photocatalytic activity.

#### 4.1.3. Crystal Facets

Surface engineering with an optimized reactive facet is a practical approach to optimizing photocatalytic activity for energy conversion and environmental remediation [83,87,101,102]. Over the past decade, significant efforts have been made to explore and understand the effect of optimizing semiconductor reactive facet in the photocatalytic activity for various applications such as water splitting [103,104,105], CO_2_ reduction [106,107], photodegradation organic pollutant [108], and antibacterial activity [109]. For instance, optimizing high exposure of reactive [001] facet from anatase TiO_2_ was shown to be effective in improving the production of H_2_ [87] and degrading organic contaminants [104]. Furthermore, Xie [110] demonstrated that a [002] dominant rectangular sheet-like WO_3_ crystal could elevate photocatalytic reduction of CO_2_ to CH_4_.

Many photocatalytic functions are improved through crystal facet engineering, such as surface energy, electronic band structure, charge transfer and separation, reactant adsorption and product desorption, and surface redox [83]. For example, the domination of exposed [001] TiO_2_ facets reduced electron holes recombination rate by capturing photoelectron and elevated surface energy, favoring photocatalytic oxidation [97]. Furthermore, Yu [111] reported that TiO_2_ contained with exposed 58% [001] and 42% [101] had the highest activity for CO_2_ photoreduction because the combination of exposed [101] and [101] facets can form surface heterojunction within a single TiO_2_. Besides favoring the separation of photo-generated carriers and enhanced photocatalytic activity [106], surface heterojunction is economically preferable since only one semiconductor is used [79].

Tailoring faceted TiO_2_ can be established by many synthesis methods, including the wet-chemistry route (hydrothermal, solvothermal, and nonhydrolytic), gas oxidation route, topotactic transformation, crystallization transformation from amorphous TiO_2_, epitaxial growth, and spray-drying [112]. However, the most used methods are hydrothermal and solvothermal due to flexibility in manipulating crystal nucleation and growth behavior [103]. Anatase TiO_2_ crystals are naturally dominated by a less reactive and stable surface [109,113]. Therefore, the key to this method is controlling crystal shape evolution during growth until desired facets can be reached using an appropriate capping agent [114].

Although many intensive experimental and theoretical methods have been studied for surface engineering of crystal facet exposure, their application has not been intensely studied in the photocatalytic conversion of methane to methanol. To our knowledge, only BiVO_4_ [86] and, recently, TiO_2_ [85] were explored for methane’s photooxidation. The remarkable results were shown by those studied. Thus, further studies investigating the crystal facet engineering of other photocatalysts concomitantly to explore the synergistic effect of other modifications such as adding H_2_O_2_, electron scavenger, and element doping will be beneficial for the photocatalytic generation of methanol.

#### 4.1.4. Defect Sites

Solar radiation has a light wavelength from 280 to 4000 nm and consists mainly of 6.6% of ultraviolet radiation (<380 nm), 44.7% of visible radiation (380–780 nm), and 48.7% of infrared radiation (>780 nm) [115]. Therefore, the photocatalyst is desired to absorb abundantly available solar energy. However, sunlight comprises light in visible and near-infrared regions with lower energy than UV light and can reduce photocatalyst performance. Thus, modification is needed to improve photocatalyst absorptivity.

Doping is a familiar modification in photocatalysts to activate poor visible light under visible-light irradiation using metal or non-metal elements. By definition, doping modification incorporates atoms or ions in a crystalline lattice, i.e., modification of the bulk structure of crystallites, but not a change of surfaces [115]. Furthermore, the doped ion can facilitate carriers to diffuse to the surface by introducing additional energy levels into the band structure that will be used to trap electrons and holes [71]. Therefore, by adding a dopant, such as iron species, in the TiO_2_, some photocatalyst properties are enhanced, including the electron-hole separation, lowering the reduction potential of H_2_O_2_ and also avoiding oxygen reduction to •O_2_^−^ [116].

### 4.2. Operational Condition

#### 4.2.1. Catalyst Loading

Loading catalyst is one of the factors that can affect the rate of photocatalytic reaction. The concentration used varies between 0.5 g/L to 10 g/L. Gondal [29] studied the effect of various catalyst loading in the photooxidation of methane to methanol in three different laser powers. It showed the dependence of methanol yield on the varied catalyst concentration (Figure 31). 

Methanol yield increased as catalyst concentration increased. In a particular concentration, it reached a maximum yield of methanol and then gradually decreased. Therefore, there is an optimum methanol yield for photocatalyst loading. Thus, an appropriate amount of photocatalyst will be effective in enhancing photocatalytic activity.

However, loading excessive catalysts will diminish the photocatalytic activity of a reaction. Such a result is believed that increasing the catalyst particle density beyond the optimum value will facilitate the recombination of electron-hole pairs [29]. Furthermore, in excessive catalyst loading, light cannot enter to activate the catalyst particle, which may scatter the photons and obstruct the light absorption [26]. Therefore, catalyst concentration substantially will increase the reaction rate, but the reaction rate will decrease [52].

#### 4.2.2. Light Wavelength and Intensity

Light is a driving force in the photocatalytic reaction to activate the photocatalyst and initiate the photocatalytic reaction. Commonly, light sources can be obtained by the conventional lamp (Xenon lamp or mercury lamp) or intense high lamp (laser lamp). However, using light from sunlight, mostly visible light, is the ultimate purpose of photocatalytic reaction. Therefore, a filter separates the UV and the Visible light from the light source in the experimental study.

Gondal [29] performed different flux densities ranging from 0.5–3 Watt, resulting in a dependency of flux density on methanol production (Figure 32a). A gradual increase in methanol yield with the flux density because a more significant amount of photons is generated to produce hydroxyl radicals. However, it decreased at a constant value after reaching the maximum point, indicating equal generation of hydroxyl radicals and recombination of electron-hole pairs. Thus, high flux density may not effectively exploit the photon generated from light since most photons escape from the reaction vessel without interaction with the catalyst. In contrast, recombination of electron-hole pairs and scattering are prominent processes for low flux density.

Shi et al. [117] performed photocatalytic methane to methanol on a fluffy metal-free carbon nitride photocatalyst in the presence of H_2_O_2_ using simulated AM1.5G and visible light >420 nm. Visible light showed a lower methane conversion than that under full sunlight (Figure 32b). Under full simulated sunlight irradiation, •OH was generated from strong photolysis of H_2_O_2_, whereas under visible light irradiation •OH was produced by photocatalytic reduction of H_2_O_2_ by photoexcited electrons (e^−^).

#### 4.2.3. Irradiation Time

The irradiation time is one of the crucial factors in the photocatalytic methane conversion to methanol. Generally, the amount of product will increase as reaction time increase since more reactant can be contacted with the catalyst. However, it does not fully apply to methanol generation from methane and water. This is because methanol, as the product, is more reactive than methane as the reactant. Thus, sufficient time will be needed to achieve the highest methanol production. 

Many studies of photocatalytic methanol production from methane, including reaction or irradiation time as a parameter to observe. Commonly, irradiation time varied between 1.5 to 8 h. Besides CH_3_OH, other side products, such as O_2_, H_2_, CO, CO_2_, and C_2_H_6_, were also observed. 

In photocatalytic methanol production from methane, the yield of methanol increased until reaching the maximum yield. Then as time increased, the yield or selectivity of methanol gradually decreased (Figure 33). It can be explained that methanol will compete with water molecules immediately after its formation for photogenerated holes, hence the degradation coincides with the generation of hydroxyl radical [32]. Another reason is reported by Gondal [29], who observed the yield of O_2_ and CH_3_OH through reaction. The decrease in methanol production could correspond to the formation of superoxide radicals (O_2_^−^) from O_2_ reduction. This superoxide radical further reacts with CH_3_OH and produces CO_2_. It is remarked by the depletion of O_2_ and the lack of methanol (Figure 33a).

#### 4.2.4. Reactor Configuration

The experiment will consist of catalyst synthesis, characterization, and photocatalytic activity study in a typical photocatalytic reaction study. A photocatalytic activity study is performed in the photoreactor, where reaction products are generated from the contact between photocatalysts, reactants, and photons. The reactor used for the photocatalytic process can be a slurry reactor, fixed bed, or membrane photoreactor [118]. The batch slurry reactor is the most frequently used reactor in the photocatalytic production of methanol from methane and water since it can get three phases (gas, liquid, and solid) contacted. In addition, the slurry reactor will provide a high surface area to illuminate and high mass transfer between catalyst and reactant through agitation. 

In designing photoreactors, there are some important factors to consider. It will influence photoreactor performance, including the light source, geometrical configuration to accommodate light, the material of construction, heat exchange, and mixing and flow characteristics [119]. The light source type determines the wavelength and intensity range used in the photocatalytic process. A good photoreactor is desired to have a high ability to collect photons that extend the area to be illuminated to achieve an increased activity in the photoreactor [120]. The material used for the photoreactor should handle temperature, pressure, and corrosion problems based on the reactant and operation condition applied. The material for the photoreactor can be quartz, ace glass, or Pyrex. The quartz photoreactor is the most frequently used in the photocatalytic reaction due to its excellent transparency range, particularly in UV light [118]. As energy is added in the form of light, heat increase is inevitable, so it is crucial to remove heat using a heat exchanger to maintain the desired temperature. Mixing characteristics is also vital in designing photoreactors to provide high contact between reactants and catalysts. The easiest method for a liquid-solid photoreactor to agitate the mixture is by using a stirrer [119].

In the photocatalytic reaction, the particle should highly absorb the light participation in the reaction to achieve a high degree of substance conversion [121]. Therefore, lamp position is one of the factors which will influence the capability of light-harvesting in the photocatalyst. In typical slurry reactors, photoreactors can be classified into several types based on their lamp position; (a) Immerse illuminated (Figure 34a), (b) Side illuminated (Figure 34b), and (c) Top illuminated (Figure 34c). Table 4 summarizes the advantages and disadvantages of each lamp position for photocatalytic reaction.

## 5. Conclusions and Future Perspective

Converting methane into higher chemical values such as methanol is an exciting prospect in the industry. Currently, methanol is produced industrially by steam reforming, which consumes high energy for syngas intermediate product and is economically not preferable. Photocatalytic technology has shown remarkable potential as a low-cost, environmentally-friendly, and sustainable technology. A thermodynamically unfavorable reaction can be performed in mild conditions using a photocatalytic reaction. The utilization of methane, water, and light which are abundant and inexpensive, to produce methanol will be a remarkable breakthrough as new energy. Many efforts have been made in this area and produced many insightful research types. However, it still encounters challenges that limit the photocatalytic reaction effectivity and application. 

Current studies on the photocatalytic conversion of methane to methanol commonly strive for a high-efficiency photocatalytic reaction (high methane conversion, methanol yield, and methanol selectivity). Selective oxidation of methane to methanol has often been considered a “holy grail” reaction in catalysis. Such a low-efficiency reaction is believed to be happened due to the stable and low polarization of methane. It leads methane to be difficult to activate and thus limits the conversion of methane. Water is used as an oxidant in a typical photocatalytic reaction of methane to methanol. Upon reaction, methane gas and an inert gas (Helium, Argon, and Nitrogen) will be purged into the photoreactor or sparged into the water for a specific duration to get a saturated mixture of water and methane. In this case, methane also has low solubility in the water (~24.4 mg/L), limiting the conversion [122]. In some studies, high pressure of methane was applied, and noticeably higher methanol yield could be achieved [27]. Since the low solubility of methane in water negatively affects the selective oxidation of methane to methanol, the study in the gas-phase condition may be considered by using humified methane [123] or oxygen as the oxidant. However, it is essential to consider that oxygen is also a more potent oxidant than water, leading to methanol overoxidation [65].

Methanol overoxidation in this reaction also becomes one of the challenges. Methanol as the product is naturally more reactive than methane as a reactant. Therefore, Overoxidation of methanol may happen when a certain amount of methanol competes with water for holes in the valence band or overoxidation by oxygen or radical superoxide [32,124]. Thus, a moderate oxidation power photocatalyst is believed to be able to prevent overoxidation of methanol such as WO_3_ [37], bipyramid BiVO_4_ [86], and {001}-dominated TiO_2_ [85]. Furthermore, adding iron species can enhance the selectivity of methanol by lowering the reduction potential of H_2_O_2_ and avoiding oxygen reduction to radical superoxide for TiO_2_ [26].

Furthermore, short reaction times by adding sufficient H_2_O_2_ should be considered to get the optimal yield of methanol [31]. To prevent methanol overoxidation, the system should be kept so that the methanol amount will not be sufficient to compete with water for holes in the valence band. Therefore, the development of gas-solid reaction [123] and the continued process should be considered a priority in future development. 

The vast majority of photocatalytic studies are one-dimensional studies that only consider the effect of one parameter. However, it can lead to bias or misinterpretation since the study cannot thoroughly investigate the impact of more than one parameter [125]. Thus, a study of the synergistic effect of some parameters is vital to prevent invisible effects for other parameters. 

Besides having precise knowledge of all relevant parameter influences in photocatalytic reaction, the kinetic model is also essential to implement this process industrially. Kinetic modeling of heterogeneous photocatalysts can be identified using Langmuir–Hinshelwood type kinetics [126]. The reaction rate by kinetic of Langmuir-Hinshelwood depends on the light intensity, which can be linear or square [29,114]. The rate is linear for low light intensity, while the rate is modeled proportionally to the square root for high intensity. The study of the kinetic model for photocatalytic conversion of methane to methanol is still very limited. Thus, the kinetic model has to be considered in the future development.

The state of the art of the current study in the photocatalytic oxidation of methane to methanol is shown in Table 3. In summary, converting methane to methanol at mild conditions photo catalytically is a promising process with high economic and environmental potential. Although many types of research have been studied in this field, this system still has many challenges. Therefore, some recommendations also have been proposed for future development. The authors hope this paper review will give a clear overview of the recent advance in the photocatalytic conversion of methane to methanol. Therefore, it can help and encourage other researchers to study this promising field. A deep and clear knowledge will trigger fast development, and, ultimately, the process can be applied in large-scale and commercial industries.

## Figures and Tables

**Figure 1 molecules-27-05496-f001:**
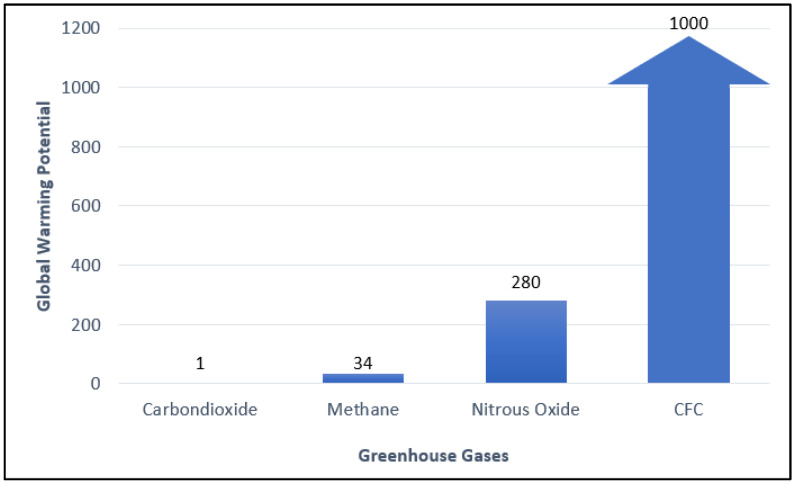
Global Warming Potential (GWP) of various types of greenhouse gases [1].

**Figure 2 molecules-27-05496-f002:**
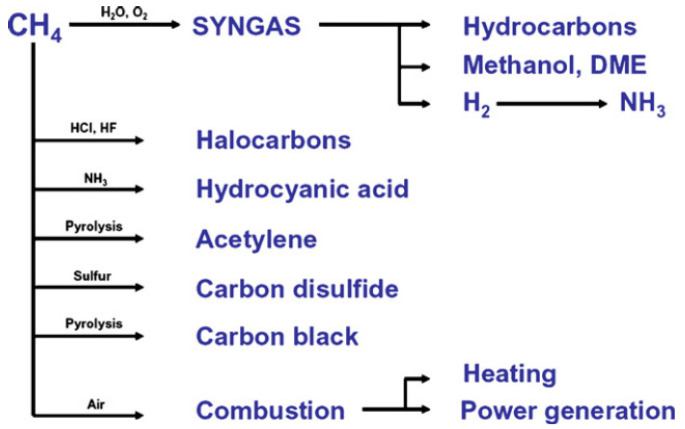
Utilization of methane in industries [3]. Reproduced with permission from Holmen. Copyright © 2022 Elsevier.

**Figure 3 molecules-27-05496-f003:**
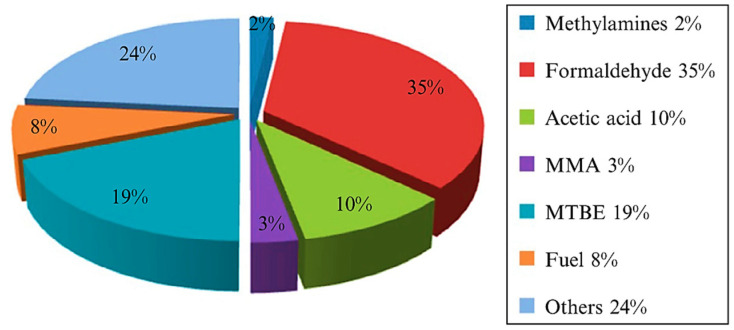
Products of Methanol in the industry [4]. Reproduced with permission from Dalena et.al. Copyright © 2022 Elsevier.

**Figure 4 molecules-27-05496-f004:**
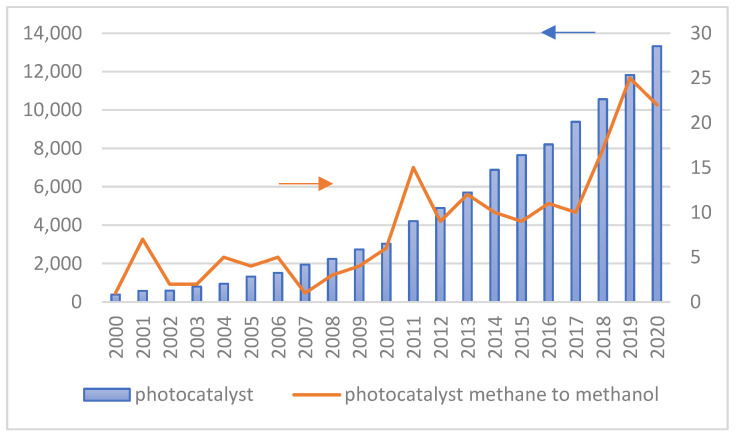
The number of research Scopus with TITLE–ABS–KEY photocatalyst–photocatalytic and photocatalyst–methane–methanol.

**Figure 5 molecules-27-05496-f005:**
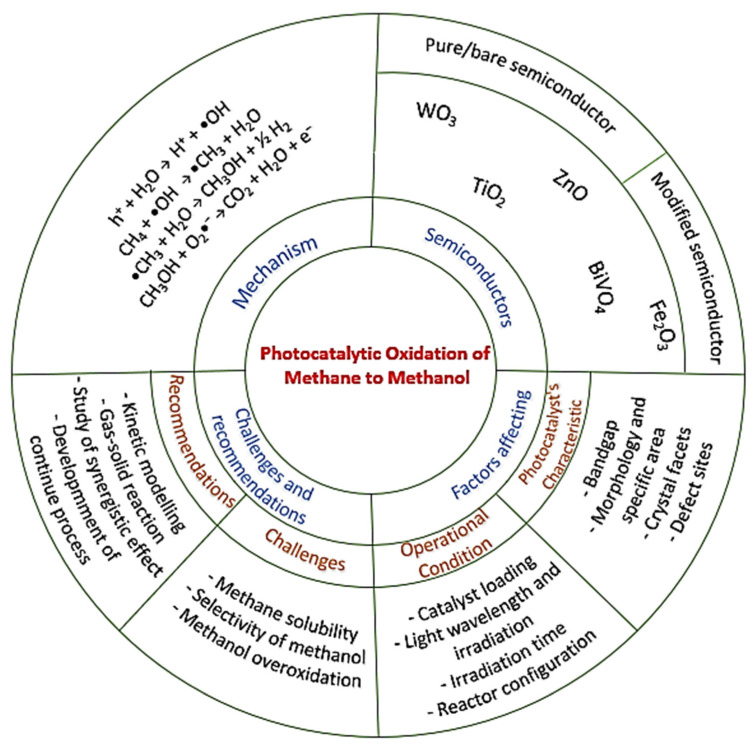
Schematic overview of photocatalytic oxidation of methane to methanol in this review paper.

**Figure 6 molecules-27-05496-f006:**
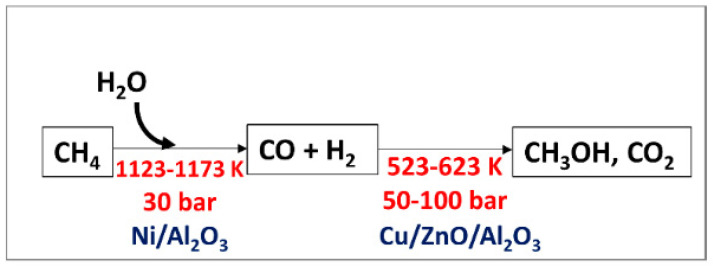
Indirect process of methanol production (Sharma et al., 2020) [5]. Reproduced with permission from Sharma et al. Copyright © 2022, MDPI.

**Figure 7 molecules-27-05496-f007:**
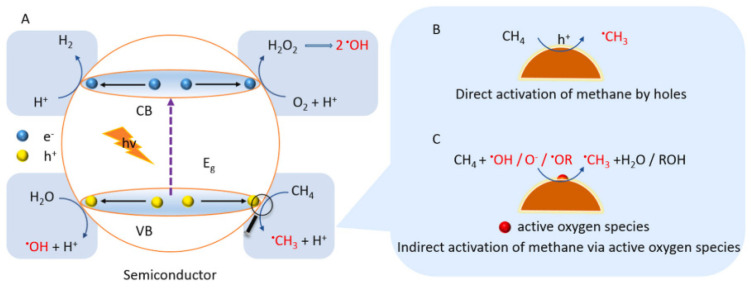
Schematic mechanism of methane activation by semiconductor [18]. Reproduced with permission from Lin et al. Copyright © 2022 Elsevier.

**Figure 8 molecules-27-05496-f008:**
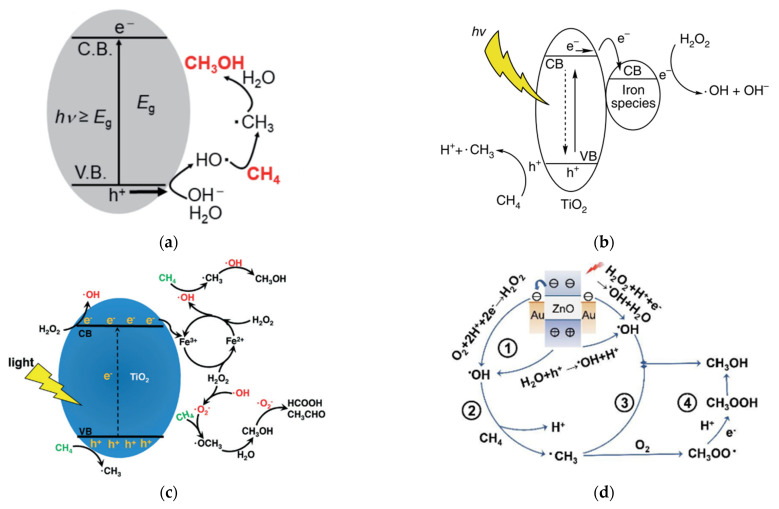
Several proposed schematic mechanisms for selective conversion of methane to methanol were reported by (**a**) Negishi et al. [25], (**b**) Xie et al. [26], (**c**) Zeng et al. [27], and (**d**) Zhou et al. [28]. Reproduced with permission from Negishi et al., Copyright © 2022 ASM Journals, Xie et al., Copyright © 2022 Springer Nature, Zeng et al., Copyright © 2022 The Royal Society of Chemistry, and Zhou et al. Copyright © 2022 The Royal Society of Chemistry.

**Figure 9 molecules-27-05496-f009:**
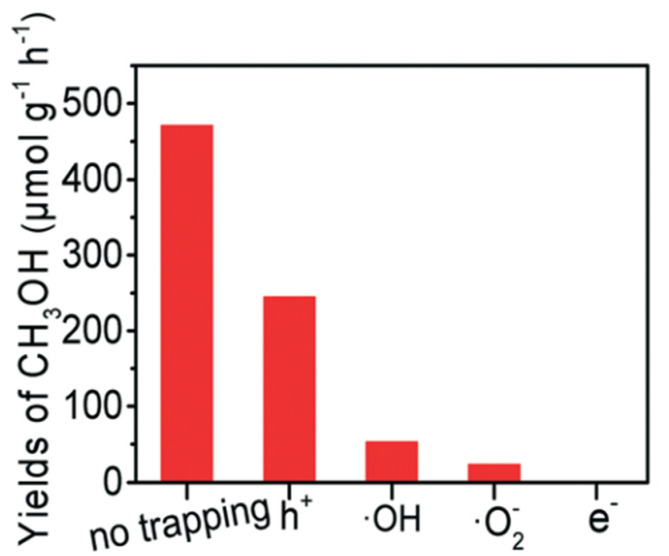
Trapping experiment of active species [23]. Reproduced with permission from Zeng et al. Copyright © 2022 The Royal Society of Chemistry.

**Figure 10 molecules-27-05496-f010:**
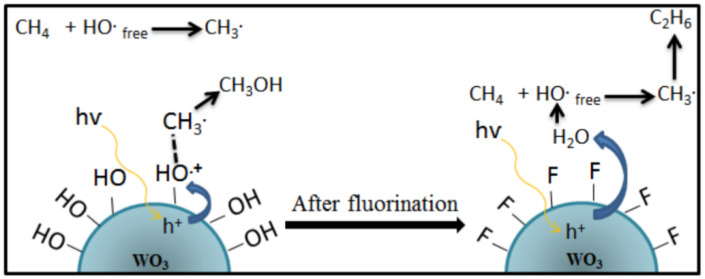
The reaction mechanism of photocatalytic conversion of methane to methanol on the surface of the WO_3_ catalyst after fluorination [24]. Reproduced with permission from Villa et al. Copyright © 2022 Elsevier.

**Figure 11 molecules-27-05496-f011:**
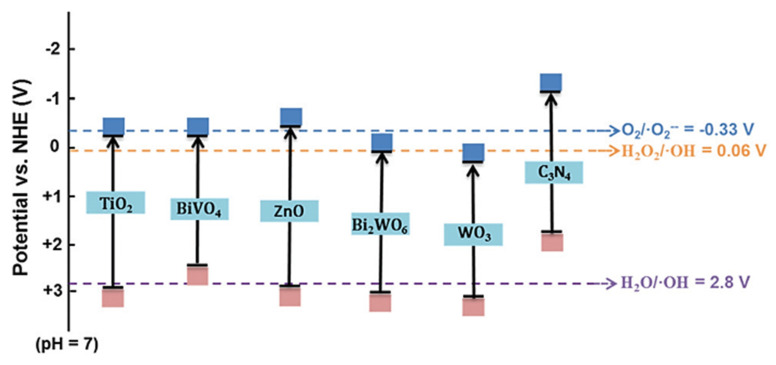
The valence band and conduction band position of possible photocatalyst used in methane conversion to methanol [35]. Reproduced with permission from Tian et al. Copyright © 2022 The Royal Society of Chemistry.

**Figure 12 molecules-27-05496-f012:**
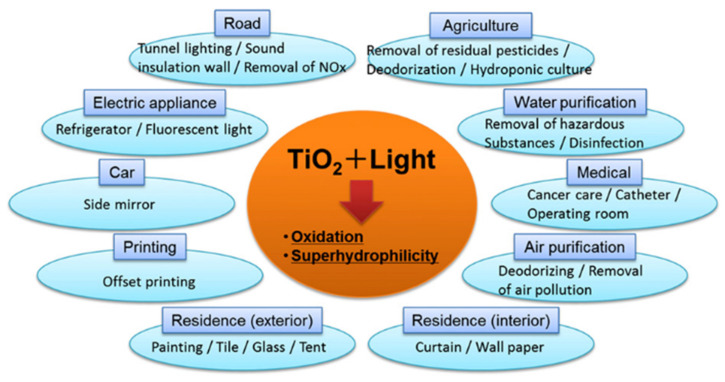
Application of TiO_2_ Photocatalyst [42]. Reproduced with permission from Nakata et al. Copyright © 2022 Elsevier.

**Figure 13 molecules-27-05496-f013:**
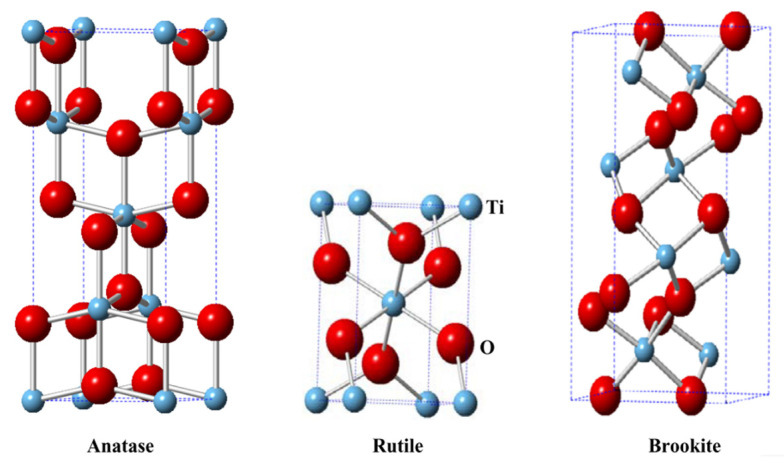
Crystal structure of anatase, rutile, and brookite [43]. Reproduced with permission from Etacheri et al. Copyright © 2022 Elsevier.

**Figure 14 molecules-27-05496-f014:**
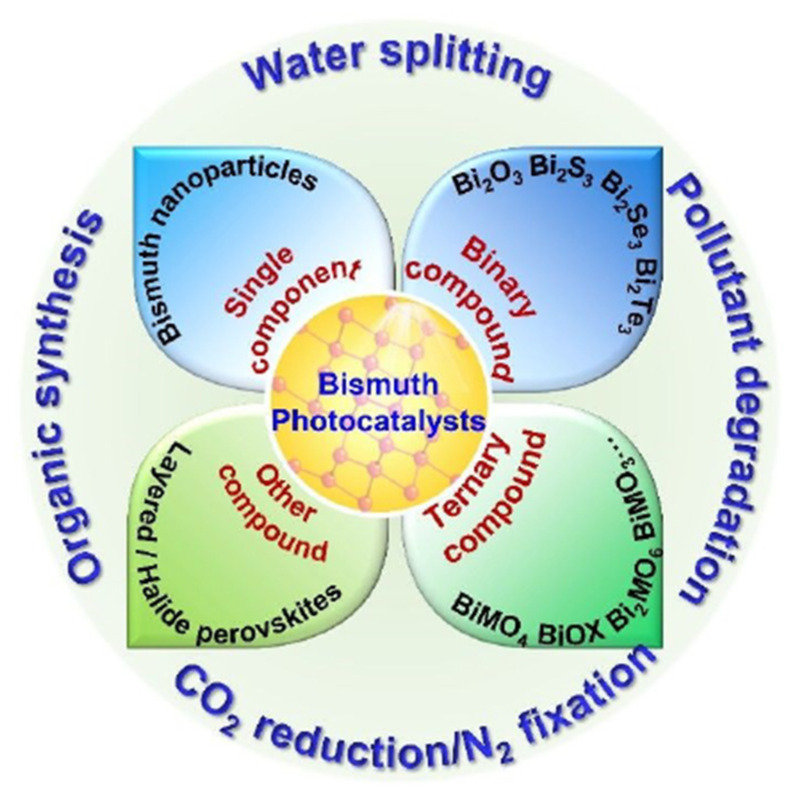
Application of bismuth-based photocatalyst [50]. Reproduced with permission from Wang et al. Copyright © 2022 The Royal Society of Chemistry.

**Figure 15 molecules-27-05496-f015:**
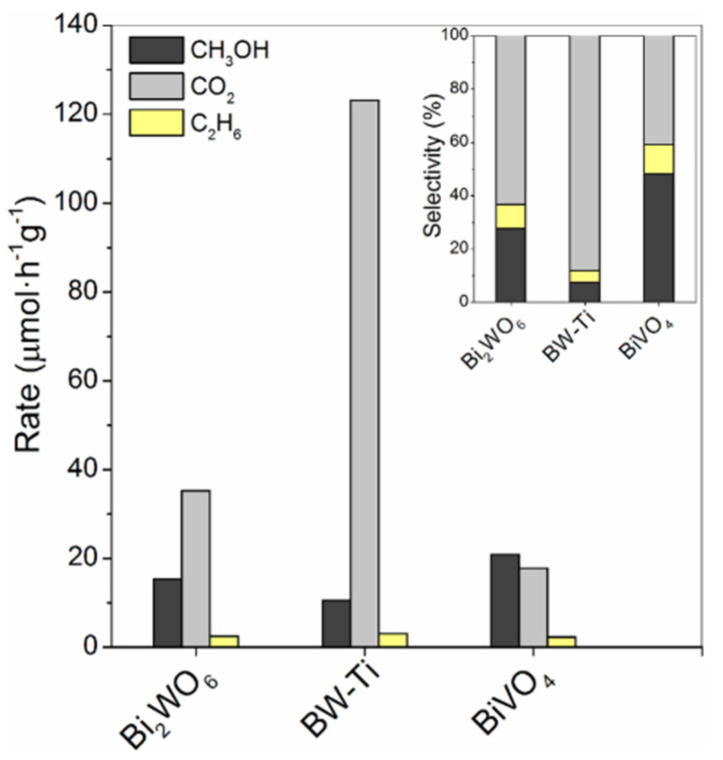
Photocatalytic activity of several bismuth-based photocatalysts [65]. Reproduced with permission from López et al. Copyright © 2022 ACS Publications.

**Figure 16 molecules-27-05496-f016:**
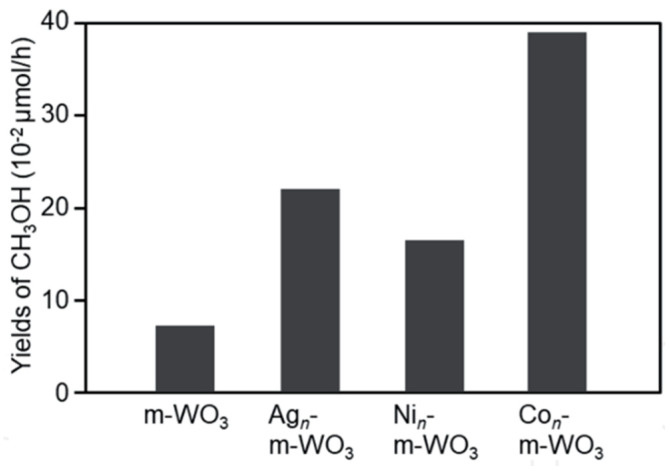
Methanol yield in several doped m-WO_3_ [25]. Reproduced with permission from Negishi et al. Copyright © 2022 ASM Journals.

**Figure 17 molecules-27-05496-f017:**
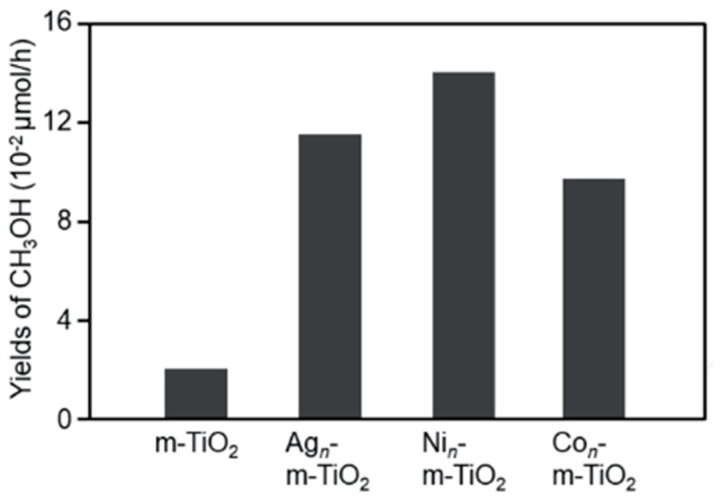
Methanol yield in several doped m-TiO_2_ [25]. Reproduced with permission from Negishi et al. Copyright © 2022 ASM Journals.

**Figure 18 molecules-27-05496-f018:**
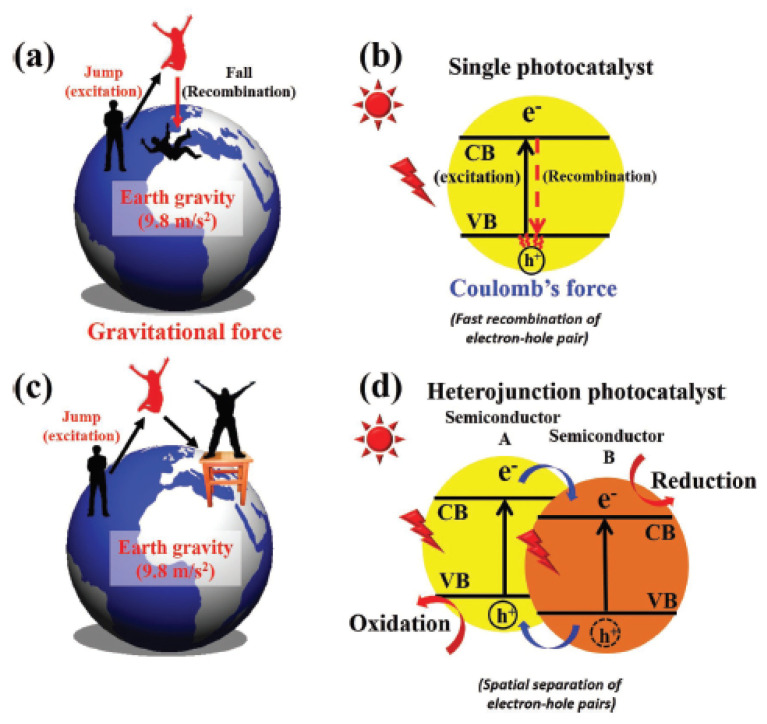
Heterojunction scheme (**a**) Fall back to the surface due to gravitational force, (**b**) recombination electron due to Coulomb force, (**c**) additional bench prevents to fall back in surface, (**d**) additional conduction band prevents to electron recombination [79]. Reproduced with permission from Low et al. Copyright © 2022 John Wiley & Sons, Inc.

**Figure 19 molecules-27-05496-f019:**
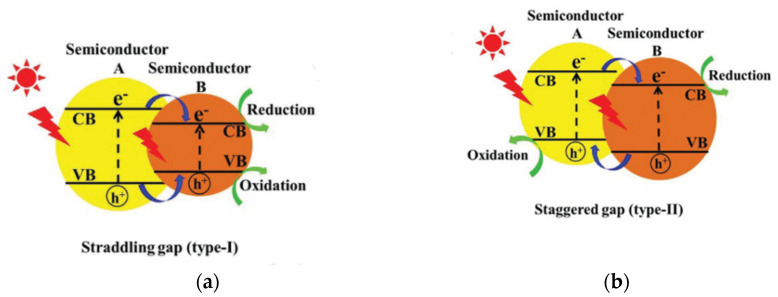
Several types of heterojunction (**a**) Type I; (**b**) Type II; (**c**) Type III; (**d**) Type p-n; (**e**) Type surface heterojunction [79]. Reproduced with permission from Low et al. Copyright © 2022 John Wiley & Sons, Inc.

**Figure 20 molecules-27-05496-f020:**
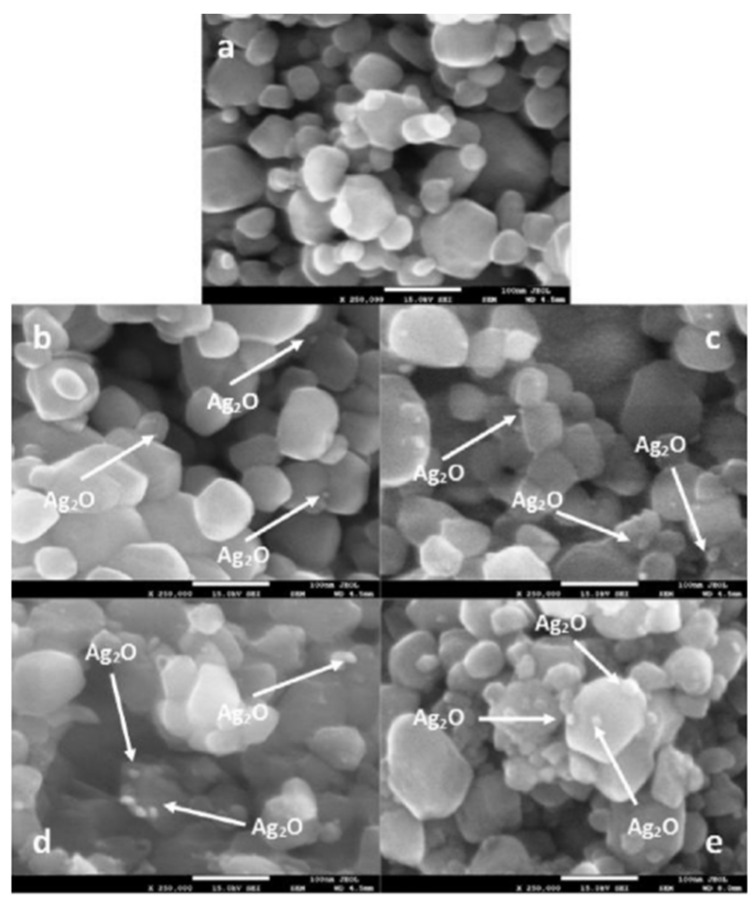
FE–SEM of (**a**) pure WO_3_ and (**b**–**e**) Ag^+^ impregnated WO_3_ [32]. Reproduced with permission from Hameed et al. Copyright © 2022 Elsevier.

**Figure 21 molecules-27-05496-f021:**
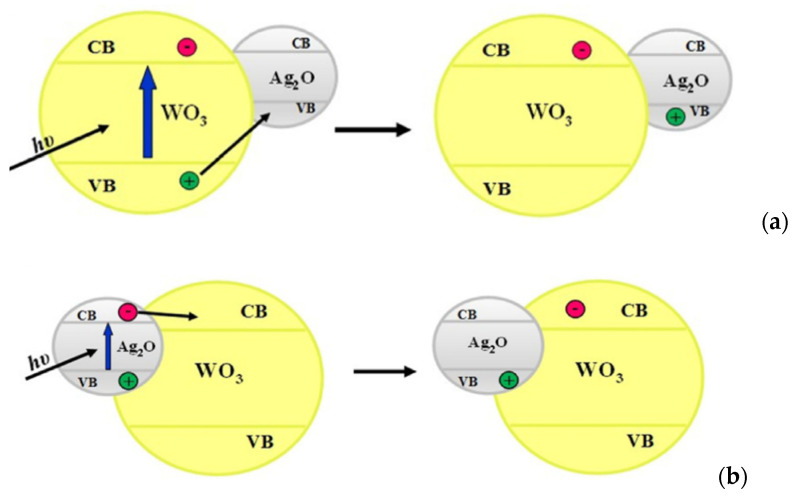
Mechanism of electron-hole pair recombination inhibition. (**a**) When WO_3_ absorbs the photons, the photogenerated holes (h^+^) are transferred from the valence band of WO_3_ to the valance band of surface Ag_2_O in energetically allowed transition. (**b**) When the photons are absorbed by surface Ag_2_O, the photogenerated electrons (e^−^) from the conduction band of Ag_2_O are transferred to the conduction band of WO_3_ [32]. Reproduced with permission from Hameed et al. Copyright © 2022 Elsevier.

**Figure 22 molecules-27-05496-f022:**
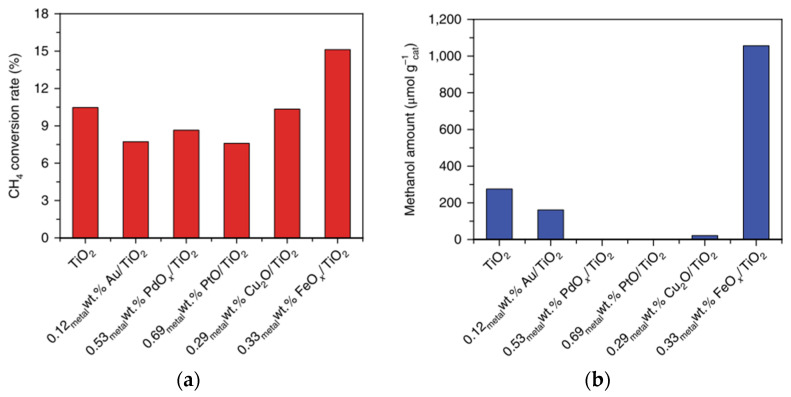
Photocatalytic activity of TiO_2_ with several co-catalysts (**a**) methane conversion and (**b**) ethanol selectivity [81]. Reproduced with permission from Xie et al. Copyright © 2022 Springer Nature.

**Figure 23 molecules-27-05496-f023:**
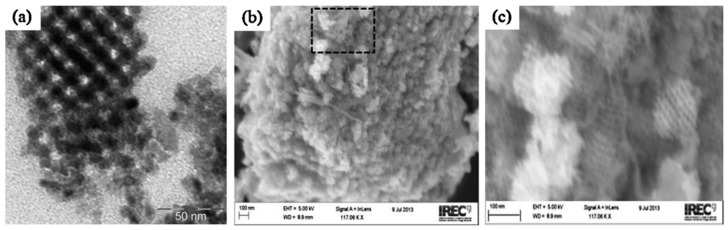
(**a**) TEM image of mesoporous WO_3_, (**b**) SEM image of mesoporous WO_3_, and (**c**) magnification of the area marked by the dotted line square in (**b**) [37]. Reproduced with permission from Villa et al. Copyright © 2022 Elsevier.

**Figure 24 molecules-27-05496-f024:**
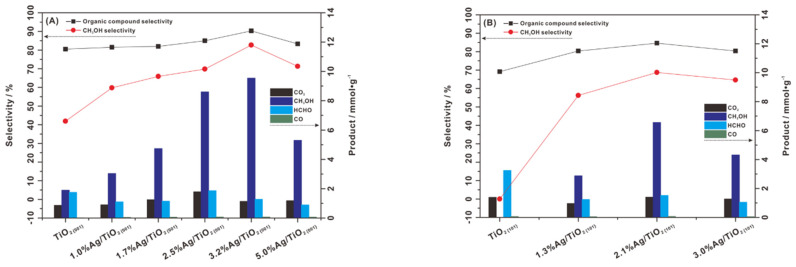
Methanol selectivity for (**A**) {001} dominated and (**B**) {101} dominated TiO_2_ [85]. Reproduced with permission from Feng et al. Copyright © 2022 Springer Nature.

**Figure 25 molecules-27-05496-f025:**
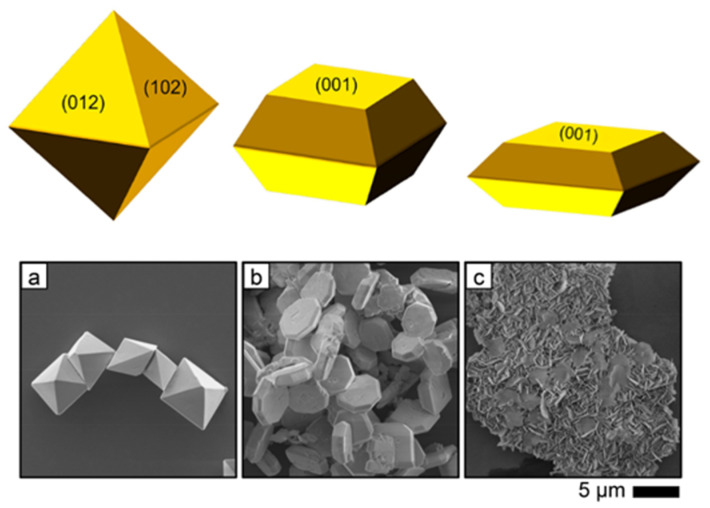
BiVO_4_ synthesized (**a**) bipyramid, (**b**) thick platelets, and (**c**) thin platelet [87]. Reproduced with permission from Wang et al. Copyright © 2022 John Wiley & Sons, Inc.

**Figure 26 molecules-27-05496-f026:**
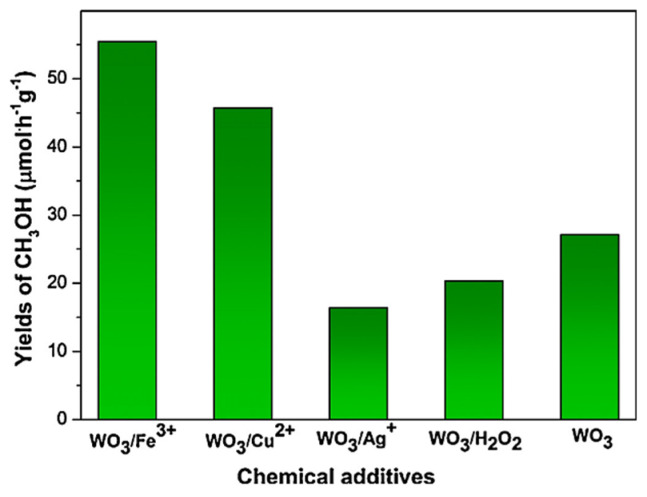
Yields of CH_3_OH in the photocatalytic oxidation of CH_4_ in systems of different electron scavengers [88]. Reproduced with permission from Villa et al. Copyright © 2022 Elsevier.

**Figure 27 molecules-27-05496-f027:**
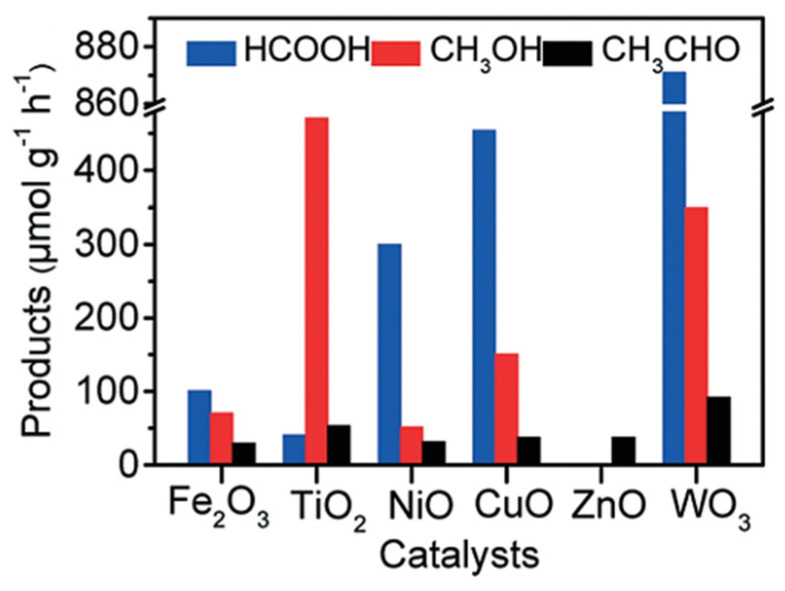
Methanol generation for different catalysts in Fenton reaction [27]. Reproduced with permission from Zeng et al. Copyright © 2022 The Royal Society of Chemistry.

**Figure 28 molecules-27-05496-f028:**
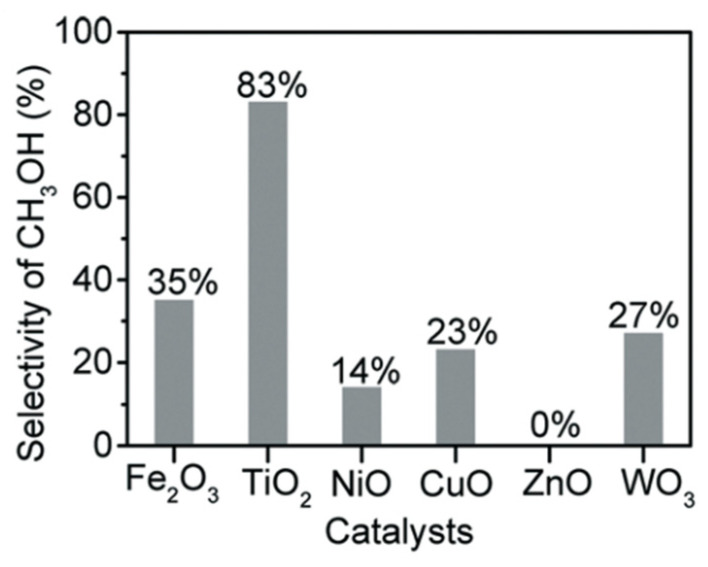
Methanol selectivity for different semiconductors [27]. Reproduced with permission from Zeng et al. Copyright © 2022 The Royal Society of Chemistry.

**Figure 29 molecules-27-05496-f029:**
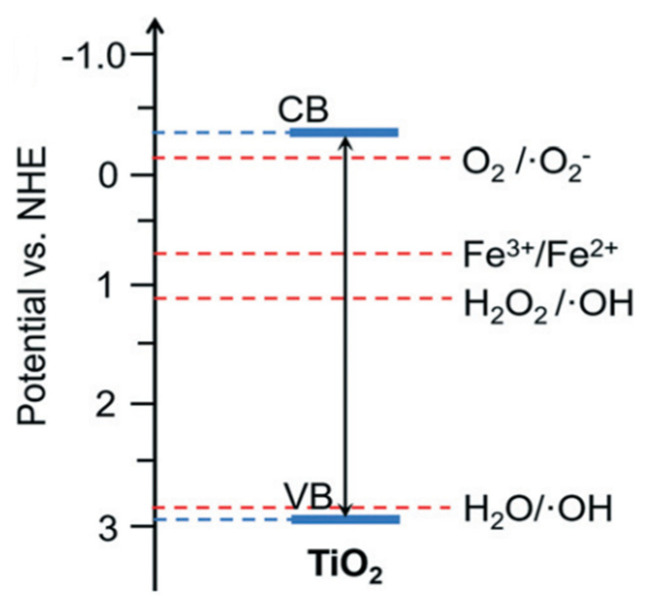
Potential redox for species generated in Fenton reaction [27]. Reproduced with permission from Zeng et al. Copyright © 2022 The Royal Society of Chemistry.

**Figure 30 molecules-27-05496-f030:**
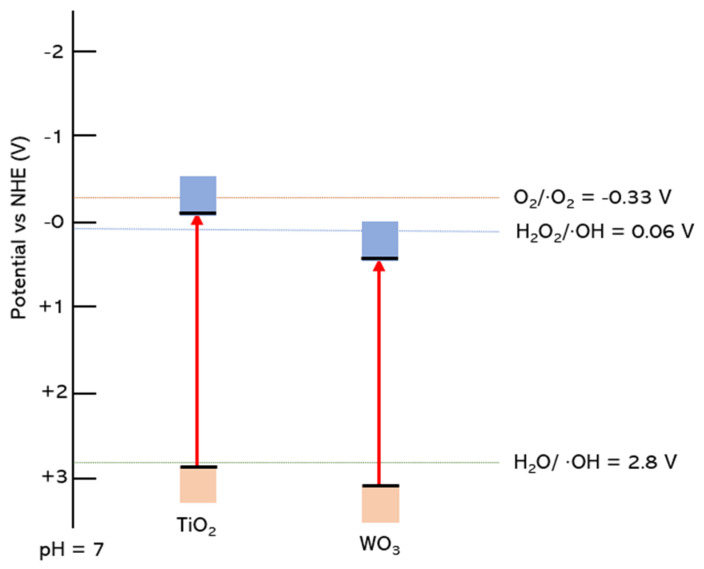
The valence and conduction band of WO_3_ and TiO_2_ photocatalyst and the potential standard reduction of hydroxyl radical and superoxide radicals.

**Figure 31 molecules-27-05496-f031:**
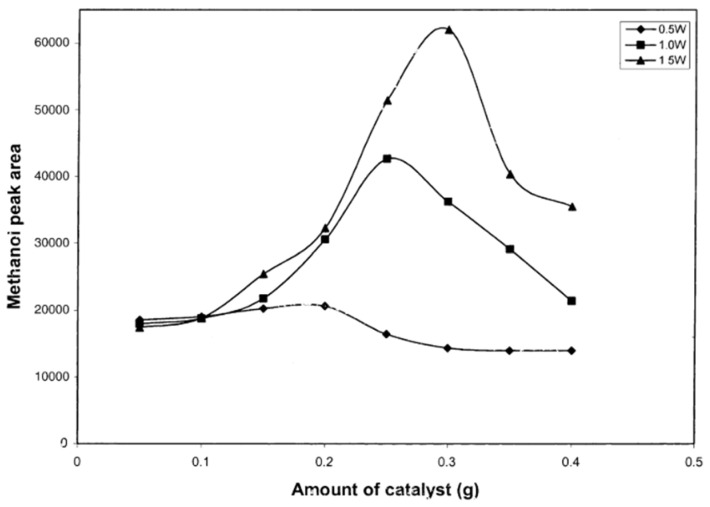
Methanol yield vs. concentration of photocatalyst [29]. Reproduced with permission from Gondal et al. Copyright © 2022 Elsevier.

**Figure 32 molecules-27-05496-f032:**
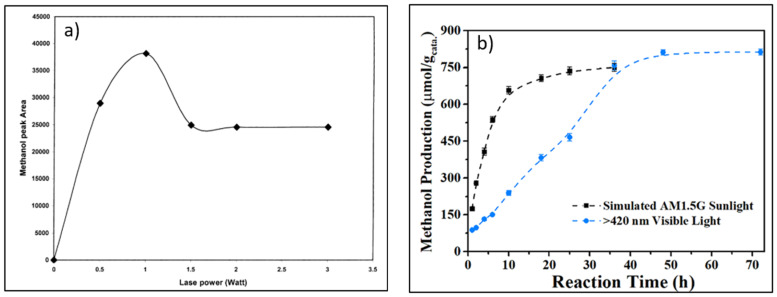
Methanol production in different light sources was reported by (**a**) Gondal et al. [29] and (**b**) Shi et al. [117]. Reproduced with permission from Gondal et al. Copyright © 2022 Elsevier and Shi et al. Copyright © 2022 John Willey & Sons, Inc.

**Figure 33 molecules-27-05496-f033:**
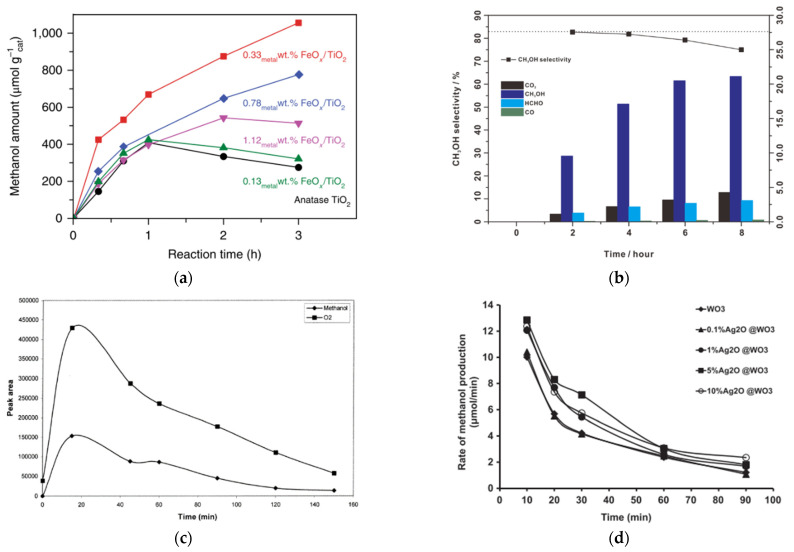
Dependency of time irradiation on production or selectivity of methanol was reported by (**a**) Gondal et al. [29], (**b**) Hameed et al. [32], (**c**) Xie et al. [26], and (**d**) Feng et al. [85]. Reproduced with permission from Gondal et al. Copyright © 2022 Elsevier, Hameed et al. Copyright © 2022 Elsevier, Xie et al. Copyright © 2022 Springer Nature, and Feng et al. Copyright © 2022 Springer Nature.

**Figure 34 molecules-27-05496-f034:**
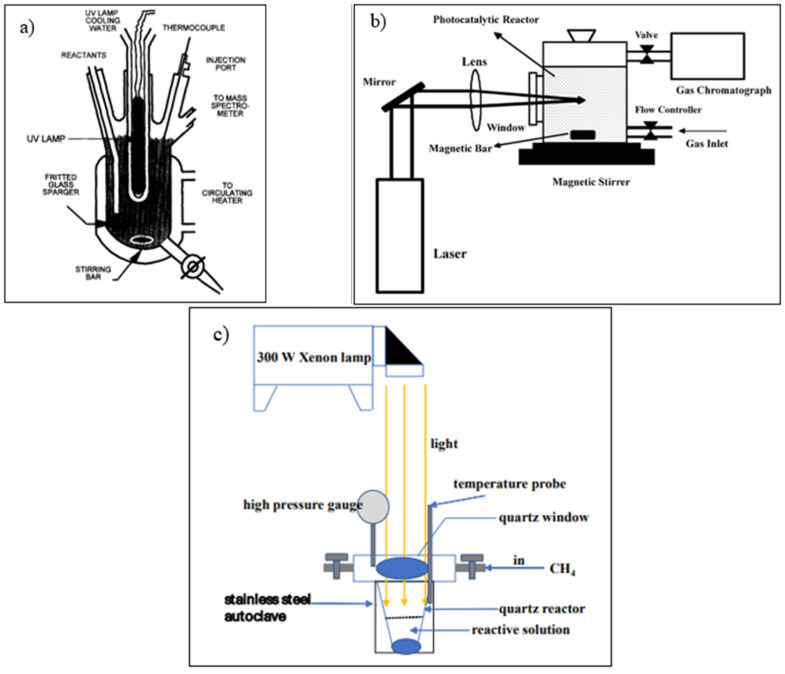
Reactor configuration with different lamp position (**a**) immersed illumination [21]; (**b**) side illumination [29]; (**c**) top illumination [27]. Reproduced with permission from Noceti et al. Copyright © 2022 Elsevier, Gondal et al. Copyright © 2022 Elsevier, and Zeng et al. Copyright © 2022 The Royal Society of Chemistry.

**Table 1 molecules-27-05496-t001:** Change of Gibbs free energy for various methane reactions [18]. Reproduced with permission from Yuliati & Yoshida. Copyright © 2022 The Royal Society of Chemistry.

No	Reaction	Chemical Reaction Equation	ΔG^0^ 298 (kJ/mol)
1	Pyrolysis	$CH_{4}\to C+2H_{2}$	50.7
2	Non-oxidative coupling of methane (NOCM)	$2CH_{4}\to C_{2}H_{6}+H_{2}$	68.6
3	Aromatization	$6CH_{4}\to C_{6}H_{6}+9H_{2}$	434
4	Total oxidation	$CH_{4}+2O_{2}\to CO_{2}+2H_{2}O$	−801
5	Oxidative coupling of methane (OCM)	$4CH_{4}+O_{2}\to 2C_{2}H_{6}+2H_{2}O$	−320
		$2CH_{4}+O_{2}\to C_{2}H_{4}+2H_{2}O$	−288
6	Partial oxidation of methane (POM)	$2CH_{4}+O_{2}\to 2CH_{3}OH$	−223
7	POM	$2CH_{4}+O_{2}\to 2CO+4H_{2}$	−173
8	POM	$2CH_{4}+O_{2}\to 2HCHO+2H_{2}$	−104
9	Methane to methanol	$CH_{4}+H_{2}O\to CH_{3}OH+H_{2}$	117
10	Steam reforming of methane (SRM)	$CH_{4}+H_{2}O\to CO+3H_{2}$	142
11	Water-gas shift reaction	$CO+H_{2}O\to CO_{2}+H_{2}$	−28.6
12	SRM + water-gas shift reaction	$CH_{4}+2H_{2}O\to CO_{2}+4H_{2}$	114
13	Methane to acetic acid	$CH_{4}+CO_{2}\to CH_{3}COOH$	71.1
14	Methane to acetone	$2CH_{4}+CO_{2}\to CH_{3}COCH_{3}+H_{2}O$	115
15	CO_2_ (dry) reforming of methane (DRM)	$CH_{4}+CO_{2}\to 2CO+2H_{2}$	171
16	Methane to amino acids	$2CH_{4}+NH_{3}+2H_{2}O\to H_{2}NCH_{2}COOH+5H_{2}$	204

**Table 2 molecules-27-05496-t002:** Free energy Gibbs of CH_4_ + H_2_O → CH_3_OH + H_2_ at various temperature.

Temperature (K)	$\Delta G^{{}^{\circ}}$
273	108
500	132
750	158
1000	183
1250	207
1500	232

**Table 3 molecules-27-05496-t003:** State-of-the-art photocatalyst development for photocatalytic conversion of methane to methanol.

No.	Photocatalyst	Synthesis Method	Photocatalyst Loading	Reactant	Light Source	Operating Condition	Reactor	Methane Conversion (X); Methanol Yield (Y); Methanol Electivity (S)	Remarks	Reference
1	La/WO_3_	Sintering	-	H_2_O/electron transfer	Mercury Lamp (222 ≤ λ ≤ 1367 nm, 46% Visible light)	P: 1 Atm T: 94 °C	Quartz photochemical reaction vessel (Immerse illumination)	X: 4% Y: n/a S: n/a	Sintered doped materials contained larger crystallites with smoother edges H_2_O_2_ improved the production of methanol	[21]
2	La/WO_3_		-	H_2_O/H_2_O_2_/ electron transfer				X: 10% Y: n/a S: n/a		
3	WO_3_	Heating H_2_WO_4_, 300 °C	0.05–0.4 g	H_2_O/H_2_O_2_/Fe^3+^	Visible laser (514 nm, 0.5 W)	P: 1 Atm T: RT	Pyrex cell (Side illumination)	X: n/a Y: 6.15 mg/L S: n/a	XPS: W:O = 1:3 H_2_O_2_ decreased the production of methanol Fe^3+^ optimized the production yield	[29]
4	WO_3_							X: n/a Y: 17.33 mg/L S: n/a		
5	WO_3_							X: n/a Y: 32.36 mg/L S: n/a		
6	WO_3_	-	5 g/L	H_2_O	UV laser beam (355 nm)	P: 1 Atm T: RT	Glass cell (Side illumination)	X: 29% Y: n/a S: n/a	Methanol degradation occurs from electron donation to valance band holes in competition with water (WO_3_) or superoxide radical oxidation (TiO_2_)	[23]
7	TiO_2_ (rutile)	-						X: 21% Y: n/a S: n/a		
8	NiO	-						X: 20% Y: n/a S: n/a		
9	Ag/WO_3_	Impregnation	~4 g/L	H_2_O	355 nm laser foton	T: RT	A self-fabricated photocatalytic reactor (Side illumination)	X: n/a Y: 12.2 µmol min^−1^ S: n/a	The increasing loading of Ag+ ions shifted the band gaps in the longer wavelength region FE-SEM: 20–40 nm diameter XPS: the more Ag^+^ ion impregnated will exist Ag_2_O on surface	[32]
10	Bi_2_WO_6_	Hydrothermal	1 g/L	H_2_O/FeCl_3_/H_2_SO_4_	Mercury lamp (450 W, λ ≥ 185 nm)	P: 1 Atm T: 55 °C	A commercial photochemical reactor—Ace Glass (Immersed illumination)	X: n/a Y: 18 μmol·g^−1^·h^−1^ S: n/a	BET: BiWO_6_ (30 m^2^ g^−1^), BW-Ti (41 m^2^ g^−1^), BiVO_4_ (2 m^2^ g^−1^) Crystallite size: BiWO_6_ (8 nm), BW-Ti (10 nm), BiVO_4_ (28 nm) BiVO_4_ showed the most selective due to moderate oxidation potential from its band edge	[65]
11	BiVO_4_							X: n/a Y: 21 μmol·g^−1^·h^−1^ S: n/a		
12	Bi_2_WO_6_/TiO_2_−P25							X: n/a Y: 15 μmol·g^−1^·h^−1^ S: n/a		
13	BiVO_4_	Hydrothermal	1 g/L	H_2_O	Mercury lamp (450 W, λ ≥ 185 nm)	P: 1 Atm T: 55 °C	A commercial photochemical reactor—Ace Glass (Immersed illumination)	X: n/a Y: 6 µmol h^−1^ S: 50%	Nitrite ion acted as •OH scavenger to prevent oxidation of CH_3_OH	[88]
14	BiVO_4_ + NO_2_^−^			H_2_O/NO				X: n/a Y: 3 µmol h^−1^ S: 90%		
15	WO_3_	Hard template	10 g/L	H_2_O	Quartz Hg-vapor lamp (λ ≥ 185 nm)	P: 1 Atm T: 55 °C	A commercial photochemical reactor—Ace Glass (Immersed illumination)	X: n/a Y: 14.5 µmol h^−1^ S: 22.0%	Methanol was produced by the reaction of hydroxyl radical in the surface	[24]
16	F/WO_3_			H_2_O/HF				X: n/a Y: 8.5 µmol h^−1^ S: 17.9%		
17	WO_3_	Hard template	10 g/L	H_2_O/H_2_O_2_	Quartz Hg-vapor lamp (λ ≥ 185 nm)	P: 1 Atm T: 55 °C	A commercial photochemical reactor—Ace Glass (Immersed illumination)	X: n/a Y: 9 µmol h^−1^ S: n/a	XRD: mWO_3_ surface area was 151 m^2^ g^−1^ Fe^3+^ and Cu^2+^ significantly improved the generation of methanol by capturing the photogenerated electrons Ag^+^ aggravated the selectivity of methanol and deposited it on the surface	[37]
18	Mesopore WO_3_/Fe^3+^			H_2_O				X: n/a Y: 55.5 μmol·g^−1^·h^−1^ S: n/a		
19	Mesopore WO_3_/Cu^2+^			H_2_O				X: n/a Y: 45 μmol·g^−1^·h^−1^ S: n/a		
20	Mesopore WO_3_			H_2_O				X: n/a Y: 25 μmol·g^−1^·h^−1^ S: n/a		
21	Mesopore WO_3_/Ag^+^			H_2_O				X: n/a Y: 15 μmol·g^−1^·h^−1^ S: n/a		
22	Mesopore WO_3_/H_2_O_2_			H_2_O				X: n/a Y: 20 μmol·g^−1^·h^−1^ S: n/a		
23	TiO_2_	-	~1 g/L	H_2_O/H_2_O_2_	Xenon lamp with 710 nm short-pass filter (300 W, 185 nm ≤ λ ≤ 710 nm)	P: 1 Atm T: RT	Custom-made reactor (Immerse illuminated)	X:10.5% Y: 290 μmol·g^−1^·h^−1^ S: n/a	XRD: Anatase crystal TiO_2_ BET: Surface area of TiO_2_ (52.38 m^2^ g^−1^), Au/TiO_2_ (42.26 m^2^ g^−1^), PdO_x_/TiO_2_ (47.49 m^2^ g^−1^), PtO/TiO_2_ (51.83 m^2^ g^−1^), Cu_2_O/TiO_2_ (45.63 m^2^ g^−1^), FeO_x_/TiO_2_ (45.52 m^2^ g^−1^)	[26]
24	Au/TiO_2_	Facile impregnation						X:7.5% Y: 170 μmol·g^−1^·h^−1^ S: n/a		
25	PdO_x_/TiO_2_	Facile impregnation						X: 8.5% Y: 0 S: n/a		
26	PtO/TiO_2_	Facile impregnation						X: 7.5% Y: 0 S: n/a		
27	Cu_2_O/TiO_2_	Facile impregnation						X: 10.5% Y: ~10 μmol·g^−1^·h^−1^ S: n/a		
28	FeO_x_/TiO_2_	Facile impregnation						X: 15% Y: 1000 μmol·g^−1^·h^−1^ S: n/a		
29	Bipyramid BiVO_4_	Hydrothermal synthesis	1 g/L	H_2_O	Xenon arc lamp (350 W, 200 nm ≤ λ ≤ 800 nm)	T: 65 °C	Custom quartz reaction vessel (Bottom illuminated)	X: 0.96% Y: 111.9 μmol·g^−1^·h^−1^ S: 85%	XRD: monoclinic scheelite structure of BiVO^4^ BET: Surface area of bipyramid (3.2 m^2^ g^−1^), thick platelet (3.6 m^2^ g^−1^), thin platelet (3.6 m^2^ g^−1^)	[86]
30	Thick platelet BiVO_4_							X: 0.72% Y: 79.2 µmol h^−1^ g^−1^ S: 85.7%		
31	Thin platelet BiVO_4_							X: 0.87% Y: 65.7 µmol h^−1^ g^−1^ S: 58.2%		
32	m-WO_3_	Hard template method	1 g/L	H_2_O	Mercury lamp (400 W, λ ≥ 185 nm)	T: 55–60 °C	Quartz glass (Immerse illuminated)	X: n/a Y: 0.08 µmol h^−1^ S: n/a	It is essential to choose an appropriate co-catalyst for photocatalyst to improve the photocatalytic activity Particle size 1 nm in this study was considered not optimal for promoting the reaction	[25]
33	Ag/m-WO_3_							X: n/a Y: 0.22 µmol h^−1^ S: n/a		
34	Ni/m-WO_3_							X: n/a Y: 0.17 µmol h^−1^ S: n/a		
35	Co/m-WO_3_							X: n/a Y: 0.39 µmol h^−1^ S: n/a		
36	m-TiO_2_							X: n/a Y: 0.02 µmol h^−1^ S: n/a		
37	Ag/m-TiO_2_							X: n/a Y: 0.12 µmol h^−1^ S: n/a		
38	Ni/TiO_2_							X: n/a Y: 0.14 µmol h^−1^ S: n/a		
39	Co/m-TiO_2_							X: n/a Y: 0.95 µmol h^−1^ S: n/a		
40	Amorphous FeOOH/m-WO_3_	Hard template method	~1 g/L	H_2_O_2_	Visible Light (400 nm ≤ λ≤ 700 nm)	P: 1 Atm T: RT	Quartz window (Top illuminated)	X: n/a Y: 211 μmol·g^−1^ h^−1^ S: 91.0%	XPS: iron species on the surface are primarily amorphous FeOOH FE-SEM: WO_3_ nanocrystals with an average size of 13 nm	[89]
41	Fe_2_O_3_	-	1 g/L	H_2_O/H_2_O_2_/FeCl_2_	Xenon lamp (300 W, 300 nm ≤ λ ≤ 2000 nm)	T: 30 °C P: 3 MPa	Autoclave (Top illuminated)	X: 0.15% Y: 100 μmol·g^−1^·h^−1^ S: 35%	An appropriate ratio of Fenton reagents (Fe^2+^ and H_2_O_2_) could contribute to the conversion of CH_4_ to selectively generate CH_3_OH	[27]
42	TiO_2_/Fe	-						X: 0.39% Y: 470 μmol·g^−1^·h^−1^ S: 84%		
43	NiO/Fe	-						X: 0.27% Y: 40 μmol·g^−1^·h^−1^ S: 14%		
44	CeO_2_/Fe	-						X: 0.43% Y: 23% S: 150 μmol·g^−1^·h^−1^		
45	ZnO/Fe	-						X: 0.05% Y: 0 S: 0		
46	WO_3_/Fe	-						X: 0.89% Y: 27% S: 350 μmol·g^−1^·h^−1^		
47	Ag/TiO_2_	Hydrothermal method	0.1 g/L	H_2_O/O_2_	Xenon lamp (300 W, 300 nm ≤ λ ≤ 2000 nm)	P: 2 MPa T: 25 °C	Batch reactor (Top illuminated)	X: 0.31% Y: 4.8 mmol g^−1^ h^−1^ S: 80%	Oxygen vacancy in [001] TiO_2_ provided a distinct intermediate and reaction pathway that improved the selectivity of methanol	[85]
48	ZnO	NaBH_4_ reduction method	0.1 g/L	H_2_O/O_2_	Xenon lamp (300 W, 300 nm ≤ λ ≤ 2000 nm)	P: 2 MPa T: 25 °C	Batch reactor (Top illuminated)	X: n/a Y: 0 S: n/a	Controlled activation of O_2_ over cocatalysts produced mild reactive oxygen species, •OOH radicals that are important in selective oxidation of methanol	[66]
49	Pt/ZnO							X: n/a Y: 85.3 µmol S: n/a		
50	Pd/ZnO							X: n/a Y: 108.2 µmol S: n/a		
51	Au/ZnO							X: n/a Y: 58.6 µmol S: n/a		
52	Ag/ZnO							X: n/a Y: 23.0 µmol S: n/a		
53	Au/AgO	NaBH_4_ reduction method	1 g/L	H_2_O/O_2_	Xenon lamp (300 W, 300 nm ≤ λ ≤ 2000 nm)	P: 15 bar T: 30 °C	Batch reactor (Top illuminated)	X: n/a Y: 1371 µmol g^−1^ S: 99.1%	CH_3_OH can be produced from the combination of •CH_3_ with either O_2_ or •OH The low-intensity density of UV light could avoid the overoxidation of methanol	[28]

**Table 4 molecules-27-05496-t004:** Advantages and disadvantages of various types of photoreactor illumination [118]. Reproduced with permission from Khan and Tahir. Copyright © 2022 Elsevier.

Light Position	Advantages	Disadvantages
Top illuminated	- Exploits higher light irradiation compared to side illumination - Good exploitation of light as when irradiations fall to the bottom of the reactor can reflect back	- The color of the solution will affect the irradiation as dark color is difficult to pass through the slurry
Side illuminated	- Easy to install - Suitable for photoreactors that do not need a deep penetration	- Uneven illumination as only one side will have higher irradiation - Good illumination will be achieved if the reactor irradiates from all sides, but this will consume high energy - The glass can scatter the light
Immerse illuminated	- Most frequently used - The higher area of illumination compares to side and top illumination	- Complex installation - The lamp in the middle of the slurry will affect the unequal mixing - The particle may stick to the inner tube - The use of sunlight will not be a possible inefficient way

## Data Availability

Not applicable.

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
