# Peer review of "Recent Advances in Photocatalytic Oxidation of Methane to Methanol"

_molecules, 2022, doi:10.3390/molecules27175496_

Round 1
Reviewer 1 Report
Please refer to the word file.

Author Response
To whom it may concern,
Please find enclosed a revised version of our manuscript entitled “Recent Advances in Photocatalytic Oxidation of Methane to Methanol” by Gita Yuniar, Wibawa Hendra Saputera, Dwiwahju Sasongko, Rino R. Mukti, Jenny Rizkiana, Hary Devianto. We appreciate the feedback we received from the reviewers and have addressed each of their comments as detailed below. Please note that changes to the manuscript based on our response are highlighted as track changes in the modified documents.

Reviewer 2 Report
In this manuscript, the authors comprehensively describe the recent progress in the photocatalytic conversion of methane to methanol. The review lists not only the results of various research, but is also presented from a critical point of view. In general, the work is written at a high scientific level. This reviewer supposes that the manuscript should be accepted for publication in Molecules after minor revision:
1. In Section 3.1, the authors focus on the most common photocatalysts for the conversion of methane to methanol. Most attention is paid to metal oxide photocatalysts. On the other hand, it seems that some attention should be given for metal-organic frameworks as promising photocatalysts [10.1002/advs.202001946; 10.1039/D1GC01690C; 10.1038/s41563-022-01279-1]. The references mentioned could also be included in the manuscript.
2. The authors write "large bandgap" (for instance, lines 525 and 804). I believe that it is more correct to say "wide band gap" or "large band gap value".
3. In Section 3.2.1, metal doping strategy for improving photocatalytic activity is discussed. It is necessary to indicate the disadvantages of noble metal cocatalysts (e.g., their high cost).
4. In Section 3.2.2, the authors describe an heterojunction approach in the photocatalytic oxidation of methane. But they do not provide the main heterojunctions which could be formed (type-I, II-, III, Z-scheme, and S-scheme). It is necessary to highlight these types of heterostructures with their advantages and disadvantages (see please for consulting and citing: 10.1002/adma.201601694;10.1002/jctb.7091).
Author Response

(The authors gave the same response as above.)

Reviewer 3 Report
The article ‘’ Recent Advances in Photocatalytic Oxidation of Methane to Methanol’’ is interested review article, however, it require major changes. Moreover the technical, new addition of the paper is very week. Following are the comments for the authors to improve the article:
11. Figure 1 in the introduction section should have reference.
22. Table 1 is very general information. It should be removed, if any important information is required it should be quoted in the text.
33. Section ‘’ Basics of Heterogeneous Photocatalytic Processes’’ itself describe the basic as in the heading title as well. Please note that this is a research review article, this kind of very basic wording should be avoided.
44. Accordingly, the article should be reviewed on the basis of new addition instead of the basic information. This should be a book chapter then.
55. Similarly, figure 6 is general information, available in the literature. Authors need to consider the definition of a critical review article, instead of just presenting the literature, which is already well available.
66. Again, figure 7 is ‘’ General schematic’’ and hence should be some modification / re-draw etc. General information is not required in any critical review article.
77. Authors need serious rework in general on whole section 2 or the paper should be considered for a book chapter.
88. Section 3 is well written, however, authors should make some synchronization of the section. It seems missing connections.
99. Authors should add a separate section of techno economic analyses of Photocatalytic Oxidation of Methane to Methanol. This should be mandatory for the success of this review article.
110. Authors should consider removal of old reference / literature. Please review it for the whole article.
Author Response

(The authors gave the same response as above.)

Round 2
Reviewer 3 Report
Accept